# Observations of extreme wave runup events on the U.S. Pacific Northwest coast

Chuan Li[1,3], H. Tuba Özkan-Haller[2,3], Gabriel García-Medina[4], Robert A. Holman[2], Peter Ruggiero[2], Treena M. Jensen[5], David B. Elson[5], and William R. Schneider[5]

[1]Department of Civil and Environmental Engineering, University of California, Los Angeles, CA, USA
[2]College of Earth, Ocean, and Atmospheric Sciences, Oregon State University, Corvallis, OR, USA
[3]School of Civil and Construction Engineering, Oregon State University, Corvallis, OR, USA
[4]Coastal Sciences Division, Pacific Northwest National Laboratory, Seattle, WA, USA
[5]National Weather Service, National Ocean and Atmospheric Administration, Portland, OR, USA

**Correspondence:** Chuan Li (chuanli@ucla.edu)

**Abstract.** Extreme, tsunami-like wave runup events in the absence of earthquakes or landslides have been attributed to trapped waves over shallow bathymetry, long waves created by atmospheric disturbances, and long waves generated by abrupt breaking. These runup events are associated with inland excursions of hundreds of meters and periods of minutes. While the theory of radiation stress implies that nearshore energy transfer from the carrier waves to the infragravity waves can also lead to very large runup, there have not been observations of runup events induced by this process with magnitudes and periods comparable to the other three mechanisms. This work presents observations of several runup events in the U.S. Pacific Northwest that are comparable to extreme runup events related to the other three mechanisms. It also discusses possible generation mechanisms and shows that energy transfer from carrier waves to bound infragravity waves is a plausible generation mechanism. In addition, a method to predict and forecast extreme runup events with similar characteristics is presented.

## 1 Introduction

Wave runup (hereafter referred to simply as runup) is defined as the maximum excursion of water level at the shoreline. Runup is an important process that contributes to coastal flooding and beach and dune erosion and accretion. Very large runup events can be dangerous to beach goers in certain regions of the world, such as the Pacific Northwest (PNW) coast of the United States. This region includes the coasts of Washington, Oregon, and Northern California. Records in this region show that large runup events are the leading cause of deaths by drowning, including incidents when the runup moves logs that then contribute to the fatality (D. Elson, personal communication, 2016). Since 2005, there have been about two drowning deaths due to large runup events each year in this region (Garcia-Medina et al., 2018).

There have been numerous studies focused on improving understanding and prediction of runup using laboratory (e.g. Hunt, 1959; Battjes, 1974; Hedges and Mase, 2004; Hughes, 2004; van der Meer and Stam, 1992; Mase, 1989; Blenkinsopp et al., 2016), field (e.g. Holman, 1986; Ruggiero et al., 2001; Baldock and Holmes, 1997; Stockdon et al., 2006; Fiedler et al., 2015), and numerical methods (e.g. Fiedler et al., 2018; Garcia-Medina et al., 2017; Montoya and Lynett, 2018). Many of these studies examine the relationship between runup and beach slope and wave conditions, usually wave height and wave length. For example, Stockdon et al. (2006) produced a relationship between the 2% exceedance value of runup maxima and the beach slope, wave height, and wave length using data from several natural beaches. Some studies examine the ability of numerical models to simulate runup. For example, Fiedler et al. (2018) show that one-dimensional non-hydrostatic models can predict runup with reasonable accuracy.

Some studies have focused on infrequent runup events with very large magnitudes that are not related to earthquakes or landslides. Aside from being potentially dangerous to beach goers, these runup events are important because they erode and deposit sediments at locations not usually affected by runup (e.g. Dewey and Ryan, 2017) and can potentially damage properties and structures (e.g. Roeber and Bricker, 2015). Observations of such runup events have so far been attributed to three mechanisms: trapped waves over shallow bathymetry (e.g. Sheremet et al., 2014; Montoya and Lynett, 2018), energetic infragravity waves generated by abrupt breaking of carrier waves (Roeber and Bricker, 2015), and long waves created by atmospheric disturbances - also known as meteotsunamis (e.g. Monserrat et al., 2006; Olabarrieta et al., 2017). It has also been implied, by the theory of radiation stress, that energy transfer from carrier waves to bound infragravity waves can result in infragravity waves of very large heights (e.g. Longuet-Higgins and Stewart, 1962; Battjes et al., 2004), and can potentially lead to very large runup. However, no observed runup with magnitude comparable to those due to the other three mechanisms have been attributed to the energy transfer mechanism.

The primary aim of this work is to show for the first time, through a set of observations, that energy transfer from carrier waves to bound infragravity waves is a plausible generation mechanism of runup with magnitudes and periods comparable to those from known mechanisms that generate extreme runup. The majority of this study is based on a set of observations on the PNW coast from January 16, 2016. On this day, at least five different large runup events - some with more than a hundred meters of horizontal excursion, and all at different locations - were captured on video by beach goers. In addition, at least two runup related injury events were documented. The video footage and injuries took place along a 1000 km stretch of coastline within 5 hours of each other. Measurements from a number of instruments at various locations are analyzed. Possible generation mechanisms and comparisons to other similar observations are discussed. Lastly, a method to predict and forecast similar events is presented.

## 2   Study site

The wave climate of the PNW coast (Figure 1) is characterized by large wave heights and long wave periods, especially in the winter. For example, from 2008 to 2018 the median and 95 percentile of significant wave height for the summer month of August are 1.4 m and 2.5 m. For the winter month of January, they are 2.8 m and 5.3 m. For peak wave period they are 8.3 s

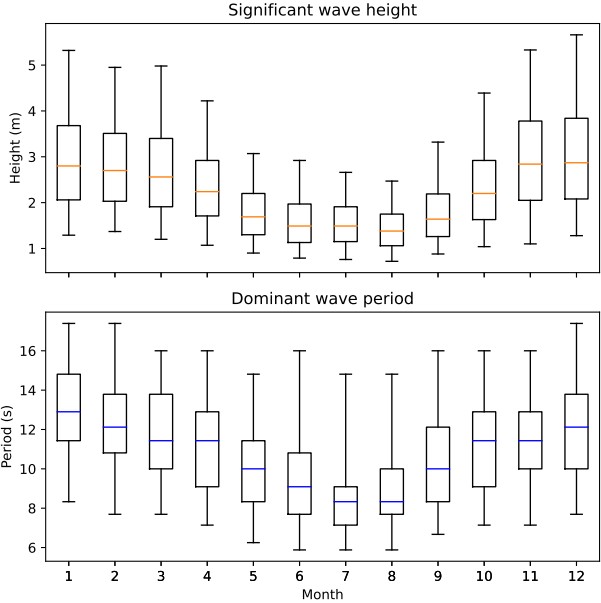

**Figure 1.** Significant wave height and peak wave period of the Pacific Northwest coasts (Washington, Oregon, and Northern California) from 2008-2018. Data includes buoys 46041, 46029, 46050, 46015, and 46022. Boxes indicate 25, 50, and 75 percentiles whereas whiskers indicate 5 and 95 percentiles (NOAA, 2020a).

and 14.8 s for August and 12.9 s and 17.4 s for January (NOAA, 2020a). The reason for this is due to the large fetch and strong winds in the north Pacific storm systems, which are especially effective during the winter as the storm systems move across the ocean basin and achieve landfall (Tillotson and Komar, 1997).

The PNW coast is also known for having low-sloping beaches. For example, upper shoreface slope from the central Oregon coast to the central Washington coast ranges between 0.005 to 0.02 (Di Leonardo and Ruggiero, 2015). There have been several studies on runup in this region. For example, Ruggiero et al. (2004) analyzed 1.5-hour water level time-series along several cross-shore transects at Agate Beach (located on central Oregon coast). During this period the offshore wave height and wave period were 2.3 m and 13 s respectively. Runup was found to vary alongshore by a factor of 2 and was found to be proportional

to foreshore beach slope. In addition, approximately 96% of the runup energy was contained in low frequencies (less than 0.05 Hz). Fiedler et al. (2015) analyzed runup on a single transect over a 44-day period, also at Agate Beach. During this time the wave height ranged from 0.5 m to 7 m. The top 2% of runup were found to be approximately linearly proportional to the square root of wave height and wave length. In addition, the amplification of runup associated with infragravity wave frequencies was found to decrease dramatically during storms. Holman and Bowen (1984) analyzed wave runup at several locations along the

mid-Oregon coast and found that 99.9% of runup variance are attributed to periods of greater than 20 s, and that 83% of runup variance are attributed to periods of greater than 50 s.

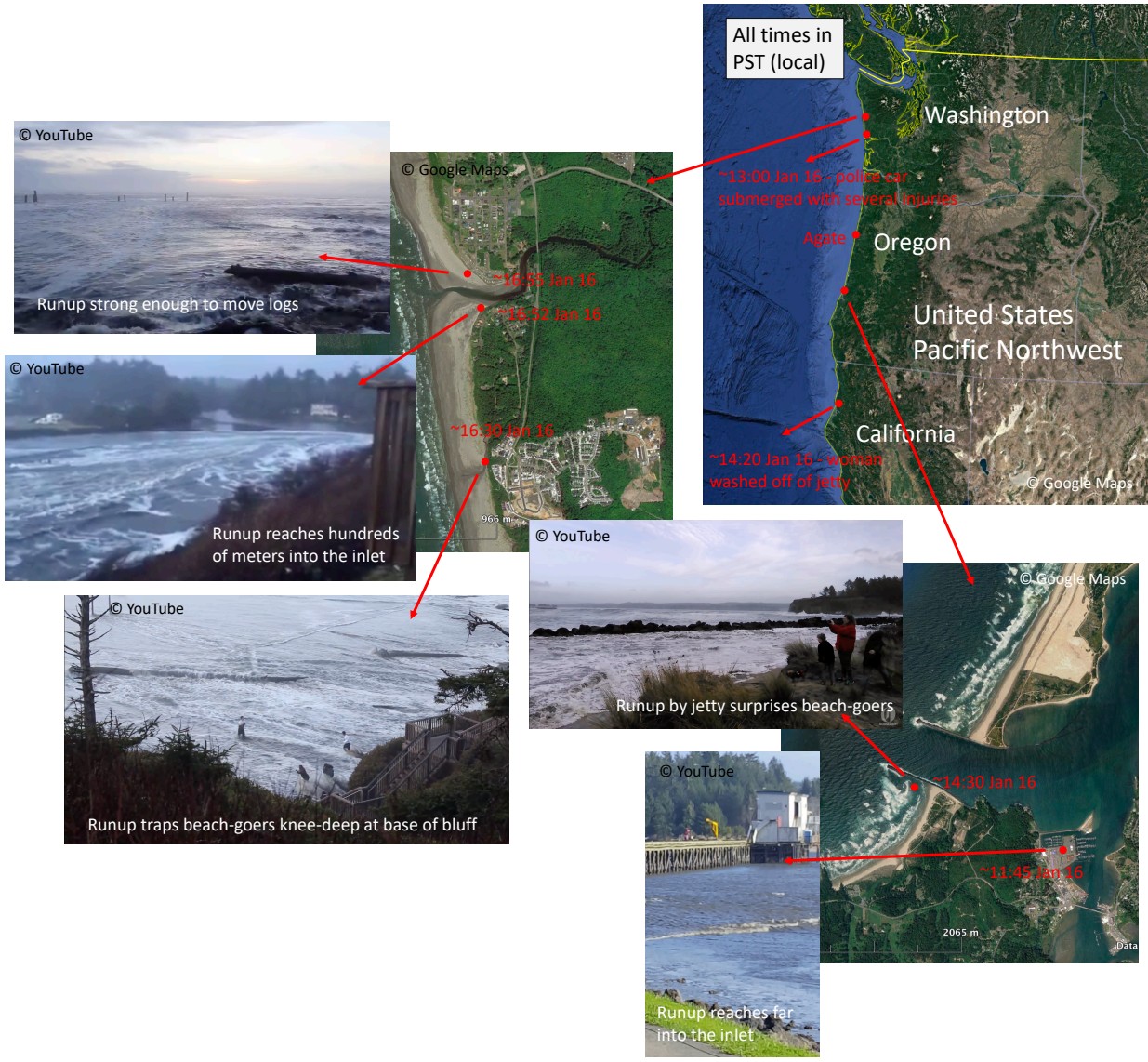

**Figure 2.** Location and approximate occurrence time of January 16, 2016 large wave runup events and stills from videos taken by bystanders. Photos are from YouTube (2020a,b,c,d,e), maps are from ©Google Maps.

## 3   Background of observed large runup events

Direct observations of unusually large runup events are scarce. A remarkably well documented case took place on January 16, 2016, when multiple unusually large runup events were observed along the coast of the PNW. There were multiple video

recordings and injury reports within approximately 5 hours and along an approximately 1000 km stretch of coastline (Figure 2). The following is a list of known large runup events in chronological order (all times in PST, i.e. local time):

– January 16 11:45, Charleston, Oregon: a video recording taken by a bystander shows a large drawdown well inside an inlet. This is followed by a succeeding large runup event at the same location approximately 3 minutes later (YouTube, 2020a).

– January 16 13:10, Ocean Shores, Washington (∼410 km north of Charleston): a report shows that around this time a police car was completely submerged by water as the police officer tried to drive away from a runup event (Jensen, 2016). Several people needed to be rescued and several were injured as they were running away from the high water event.

– January 16, 14:20, Humboldt Bay, California (∼290 km south of Charleston): a woman was washed off of a jetty and was later recovered (Jensen, 2016).

– January 16 14:30, Charleston, Oregon: an extreme runup event progressed hundreds of meters inland and surprised many beach goers, including the video taker (YouTube, 2020b).

– January 16 16:30, Seabrook, Washington (∼430 km north of Charleston): a very large runup event strong enough to move logs traped several beach goers knee-deep in water in front of bluffs (YouTube, 2020c).

– January 16 16:52, Pacific Beach, Washington (∼430 km north of Charleston): a large runup event progressed hundreds of meters inland and proceeded to travel up a small coastal stream (YouTube, 2020d).

– January 16 16:55, Pacific Beach, Washington: a large runup event mobilized several logs and pushed them against a stretch of riprap. A large reflected wave is also seen traveling offshore (YouTube, 2020e).

Large water-level fluctuations were observed along the same 1000-km stretch of coast during the same time by tide gages with both 6-minute and 1-minute recording intervals. The amplitudes of these water level fluctuations reached as high as about 0.5 m. Further detail is shown in the results sections. Wave runups of similar scale in magnitude have been observed at other times in this region, though typically not with as many video recordings from bystanders across the stretch of coastline as was on January 16, 2016.

## 4 Methods

Observations from three sources are presented in this study to provide various data across a range of locations and over different water depths. Table 1 lists the data types, measurement frequency, water depth, and distance from shore for each site. Figure 3 shows the locations of observation sites. Water level, wind speed, and atmospheric pressure from six NOAA CO-OPS stations are used. These gages are located at the coast and span approximately 800 km of coastline between Northern California and

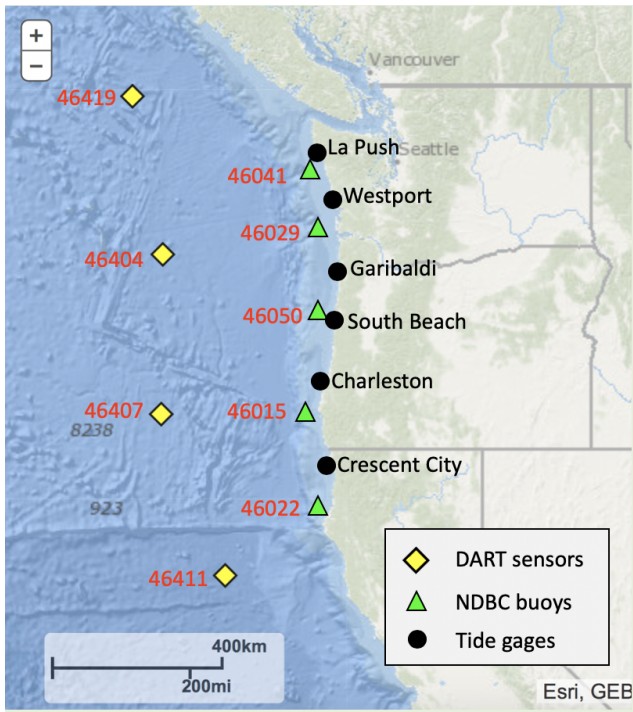

**Figure 3.** Locations of observation sites (map from NOAA, 2020a).

Northern Washington. In addition, wave height, wave period, and wave energy density spectra from five NDBC buoys are
used. These buoys span a similar range as the NOAA CO-OPS tide gages but are located further offshore on the continental
shelf. Also used are water column height from four bottom sensors from the Deep-ocean Assessment and Report of Tsunamis
(DART) system. These sensors span approximately 1200 km from north to south and are much further offshore than the NDBC
buoys and are in the deep ocean. We note that the only nearshore measurements available to us were data from tide gages. Future
studies could benefit from deployment of additional nearshore sensors. However, planning and deployment in anticipation of
similar extreme events may be challenging.

## 4.1 Tide gages

The NOAA CO-OPS tide gages (NOAA, 2020b) are located onshore. Five out of the six tide gages used in this study are located
inside estuaries (except for Crescent City which is in a harbor). Figure 4 shows satellite images around each sensor. Data used
in this study at these locations include water level, atmospheric pressure, and wind speed. Water level is measured either from
acoustic ranging (at Westport, Garibaldi, South Beach, and Charleston) or microwave radar sensors (at La Push and Crescent
City) (NOAA, 2020b). Data is sampled at 1 Hz and recorded at 1-minute intervals as 1-minute averages. Frequency analysis on
the water level time series are performed via Fast Fourier Transform with 18 degrees of freedom over record lengths between
708 and 715 samples. Atmospheric pressure is measured from pressure sensors mounted between 7.7 m to 11 m above mean

sea level at the tide gage locations (NOAA, 2020b). 21 six-second samples over 2 minutes are averaged and collected every

6 minutes. Wind speed is measured from anenometers mounted between 11 m and 30 m above sea level. 2-minute average of 1-Hz samples are collected every 6 minutes.

For the analysis sought in this work, it is important to determine the intensity of water level fluctuations at the tide gages. One method of representing such intensity is by generating an envelope of the time series. This is often done by using a Hilbert transform. When the time series has signals with high frequencies, however, the resulting Hilbert transform will also contain

high frequency signals. In this case, it is necessary then to remove the high frequency signals by a low-pass filter. Another method is to compute the root-mean-square (RMS) of the time series over a specified window. The two methods yield similar results. The RMS method is chosen over the Hilbert transform method for its simpler implementation.

## 4.2 Buoys

NDBC buoys (NOAA, 2020a) are moored buoys located 45 – 85 km offshore at water depths of 128-400 m. All stations

used in this study are of the 3-meter discus type buoys. Data used at these locations include significant wave height and peak wave period - both recorded at 1-hour intervals. Also used at these locations are wave spectral density, also recorded at 1-hour intervals, across a frequency range of 0.02 Hz to 0.485 Hz. Data acquisition starts at the 20th minute of each hour and continues for 20 minutes. During this time, buoy motions are measured and then transformed from the temporal to the spectral domain.

The energy density spectra derived from wave motions contains information of all the wave components that make up the

sea state. Useful parameters that can be computed from ocean wave spectra are the spectral moments:

$$m_n = \int_{-\infty}^{\infty} f^n E(f) df, \tag{1}$$

where $n$, typically an integer, denotes the $n$-th moment, $E(f)$ is the energy density, and $f$ is the frequency. In metric units, $E(f)$ is in m$^2$/Hz, $f$ is in Hz, and $m_n$ is therefore in m$^2$Hz$^n$. Significant wave height is approximated as as four times the square root of the zeroth moment of the wave spectra. Peak wave period is calculated as the inverse of the peak frequency.

When $n$ is taken to be negative, wave energy associated with lower frequencies is emphasized more than wave energy associated with higher frequencies. The use of negative moments has been employed by Hwang et al. (2011) to facilitate the separation of swell and wind waves. This is useful in this study as it can serve as an indicator of a swell-dominated sea-state.

## 4.3 Bottom sensors

DART sensors (DART, 2020) are located 280 – 560 km offshore at water depths of 2805-4319 m. Water column heights

are typically recorded in 15-minute intervals, although 1 min and 15 s intervals are used during special operation modes. In standard operating mode, pressure is measured at 15-second intervals and converted to water column height, but the data is only recorded every 15 minutes and transmitted every hour. When an event is detected by its tsunami detection algorithm, i.e. when the difference between water column height based on predicted tide and the measured values exceeds a threshold (30 mm in North Pacific), the instrument begins operating in event mode (DART, 2020). During event mode, 15-second values

**Table 1.** List of observations with their locations and additional information (water depth, distance to coast, data used)

| Name | Instrument | Measurements used | Measurement frequency | Water depth (m) | Distance from shore (km) |
|---|---|---|---|---|---|
| 46419 | DART bottom sensors | Water column height | 15 min, 1 min, 15 s | 2805 | 556 |
| 46404 | | | | 3738 | 426 |
| 46407 | | | | 3300 | 389 |
| 46411 | | | | 4319 | 278 |
| 46401 | NDBC buoys | Wave height, wave period, energy density spectra | 1 hour | 128 | 83 |
| 46029 | | | | 134 | 37 |
| 46050 | | | | 140 | 37 |
| 46015 | | | | 400 | 28 |
| 46022 | | | | 382 | 31 |
| La Push Westport Garibaldi South Beach Charleston Crescent City | NOAA tide gages | Water level, wind speed, atmospheric pressure | 1 min | N/A (shallow) | 0 (onshore) |

are transmitted in the initial 4 minutes and 15 seconds, followed by four hours of 1-minute averages. Afterward, the system resumes standard operation if no further events are detected.

## 5 Results

### 5.1 Observation of environmental conditions

Water level observations at the coast measured by NOAA CO-OPS tide gages in 1-minute increments are shown in Figure 5. A
set of water level fluctuations with frequencies higher than the tidal signal and magnitudes as high as 0.5 m can be seen from roughly January 16, 8:00 to January 17, 16:00 (PST, local) across all tide gages used in this study. These magnitudes of water level fluctuations are comparable to those from meteotsunamis and even some tsunamis from earthquakes (Monserrat et al., 2006; Olabarrieta et al., 2017). These fluctuations are more intense in the northern-most (La Push) and the two southern-most tide gages (Charleston and Crescent City) than the three middle tide gages (West Port, Garibaldi, and South Beach). The water
level fluctuations first increase, then are sustained around their peak level from January 16 10:00 to January 16 20:00. This period encompasses the period during which videos of large runup and injury reports took place, i.e. January 16 13:00 – January

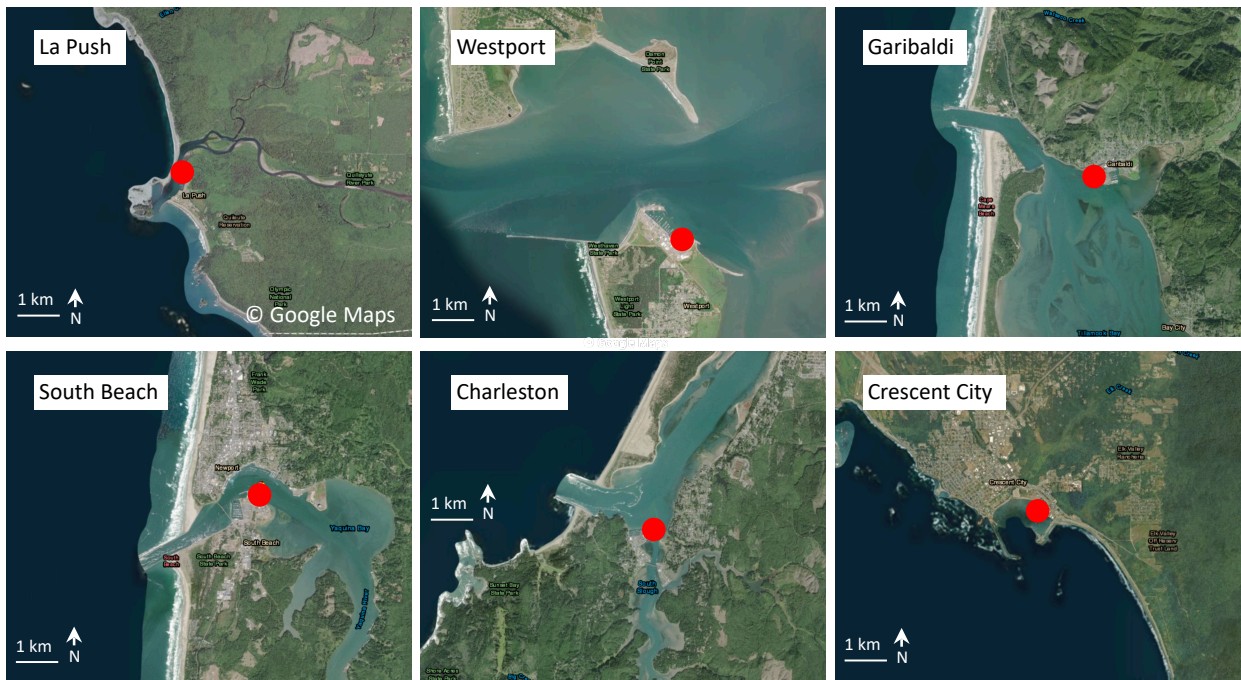

**Figure 4.** Locations of NOAA tide gages (red dots) (from NOAA, 2020b). Satellite images are from ©Google Maps.

16, 17:00. The intensity of fluctuations decreases gradually afterwards from approximately January 16 20:00 to January 17, 16:00.

Spectral analysis is performed on water level time-series from January 16 10:00 to January 16, 22:00, a period during which
intense water level fluctuations persist. An examination of the energy density spectra (Figure 6) shows the existence of a common peak period at ∼5 minutes across all tide gages (between 4.5 to 5.9 minutes). As described above, a video taken near Charleston during this time showed a lapse of approximately 3 minutes between the trough of a previous large wave runup and the crest of the following large wave runup, suggesting a runup period of approximately 6 minutes. Another spectral peak between approximately 13 to 22 minutes can also be seen for four (Westport, South Beach, Charleston, and Crescent City) of
the six stations. These periods correspond to periods of shelf resonance, and are further discussed in later sections.

Atmospheric pressure and wind speed at two tide gage locations are shown in Figure 7. Atmospheric pressure varies over a range of approximately 10 HPa between January 15 12:00 to January 17 12:00. The majority of this variation occurs over two cycles within the two days. 1-hour high-passed time series indicate that the largest high frequency anomaly in atmospheric pressure for this period is about 1 HPa. The largest high frequency wind speed anomaly over this period is about 5 m/s at South
Beach and 2.5 m/s at Charleston.

Significant wave height and peak wave period at NDBC buoys are shown in Figure 8. Wave height is seen to be moderately high for this region (∼4 to 6 m) at the approximate onset of the unusual water level fluctuations reported by the tide gages and

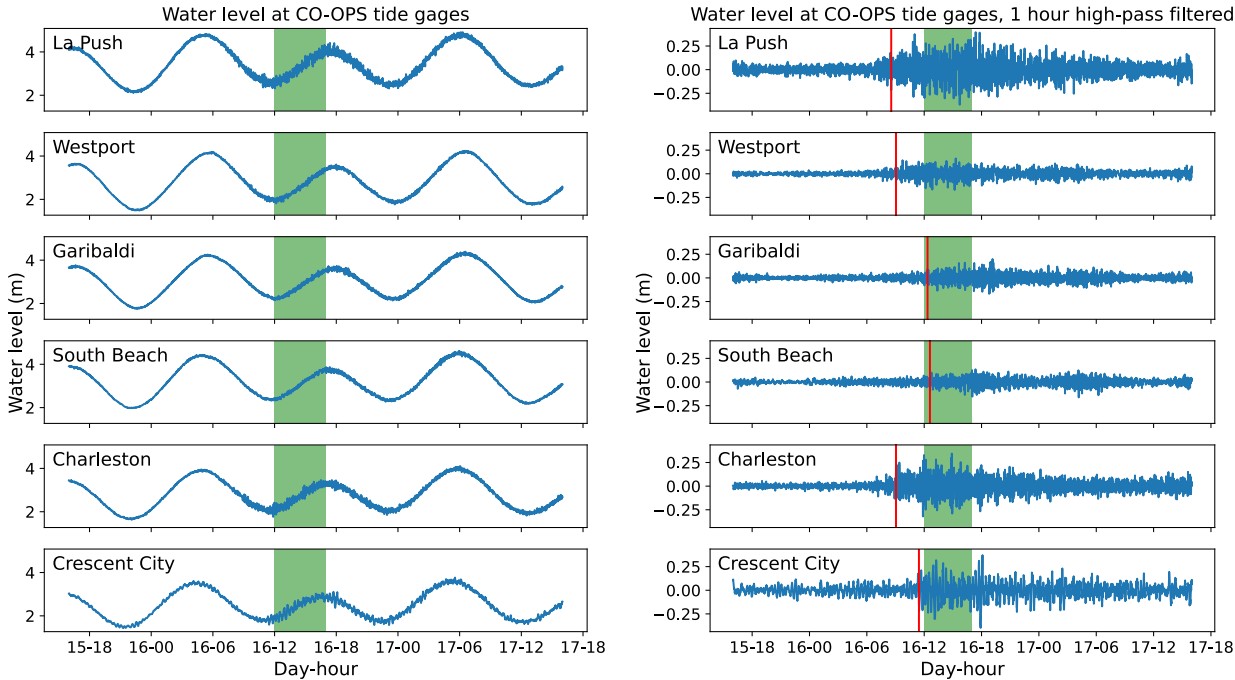

**Figure 5.** Water level at NOAA CO-OPS tide gages from January 16 – 18, 2016 (left, ordered north to south from top to bottom, time in PST, i.e. local). Green bar indicate duration of observed large runup events. (Right) one-hour high-passed version of the left. Red vertical line represents the approximate onset of the anomalous water level fluctuations, defined as having one-hour high-passed fluctuations greater than 2.5 times its standard deviation.

throughout the time period during which videos of the large wave runups and injury reports took place. No significant anomaly in wave height was observed. However, large increases of peak wave periods (from ∼12 s to ∼25 s) were observed within the recording interval of 1 hour and very close to the approximate onset of the unusual water level fluctuations at the tide gages.

Inspection of ocean wave spectra at the NDBC buoys (Figure 9 left subplot shows an example at buoy 46015) reveal a rapid and significant growth of a low frequency peak that began close to the approximate onset of the unusual water level fluctuations at the tide gages. In a further analysis, the ocean wave spectra are partitioned into a low frequency swell component and a high frequency swell and wind component at a separation frequency of 0.06 Hz, chosen to correspond to the low energy region between the two major peaks on January 16, 2016. The significant wave height is calculated at every hour for swell and wind components separately from the zeroth moments of each component. The resulting time series is plotted on Figure 9 (right subplot). It can be seen that the significant wave height for the wind component does not vary considerably during the January 16 event. However, significant wave height associated with the swell component across all 5 NDBC buoys increases by approximately 5 m over 12 hours starting close to the onset of the unusual water level fluctuations at the tide gages.

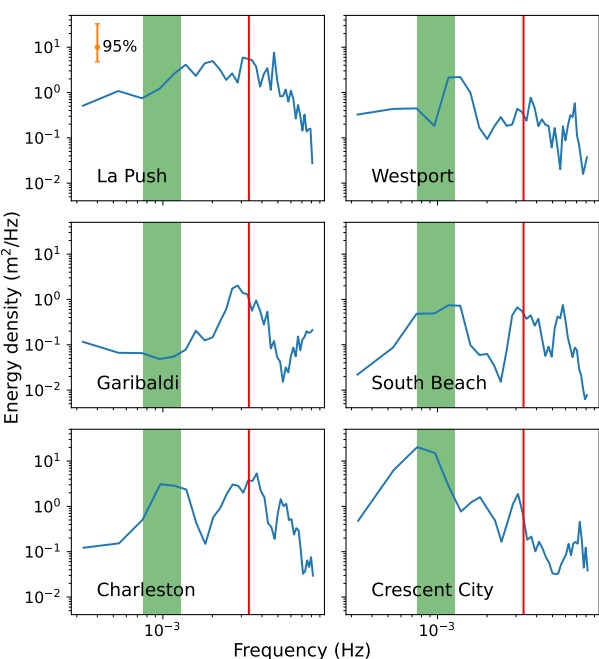

**Figure 6.** Energy density spectra of water level at NOAA CO-OPS tide gages from 2016 January 16, 10:00 to January 16, 22:00. Red vertical lines are located at periods of 5 minutes. Green box spans the periods between 13 to 22 minutes. Analysis is performed via Fast Fourier Transform with 18 degrees of freedom over record lengths between 708 to 715.

Water column height as measured by DART sensors far offshore ($280 - 560$ km, Figure 3) is shown in Figure 10. Fluctuations of higher-than-usual magnitudes are observed between about 4 hours (sensor 46407) to about 6 hours (sensor 46404) after the approximate onset of large water level fluctuations at the tide gages (i.e. January 16, 8:00). The increase in recording intervals starting near January 16 16:00 was due to transition from 'standard' to 'event' mode, which is triggered by the higher-than-usual magnitude of the water column height fluctuations.

**5.2    Possible generation mechanisms**

One possible generation mechanism for large runup involves trapped waves over shallow bathymetry (e.g. Sheremet et al., 2014), such as the continental shelf. On examination of the energy spectra of the onshore water level (Figure 6), the longer period peak (13 to 22 min) is close in magnitude to that due to resonance from the shelf in this region (Allan et al., 2012). For example, Allan et al. (2012) found that the periods of shelf resonance observed after the 2011 Tohoku tsunami were between

17 minutes and 64 minutes along the coast between Washington and Northern California. Even though there is considerable amount of energy at the ~20 min periods at some of the locations, runup characteristics from the videos on January 16, 2016 show that the period of the runup events are closer to 5 minutes than they are to 20 minutes. We also note that the two tide gages

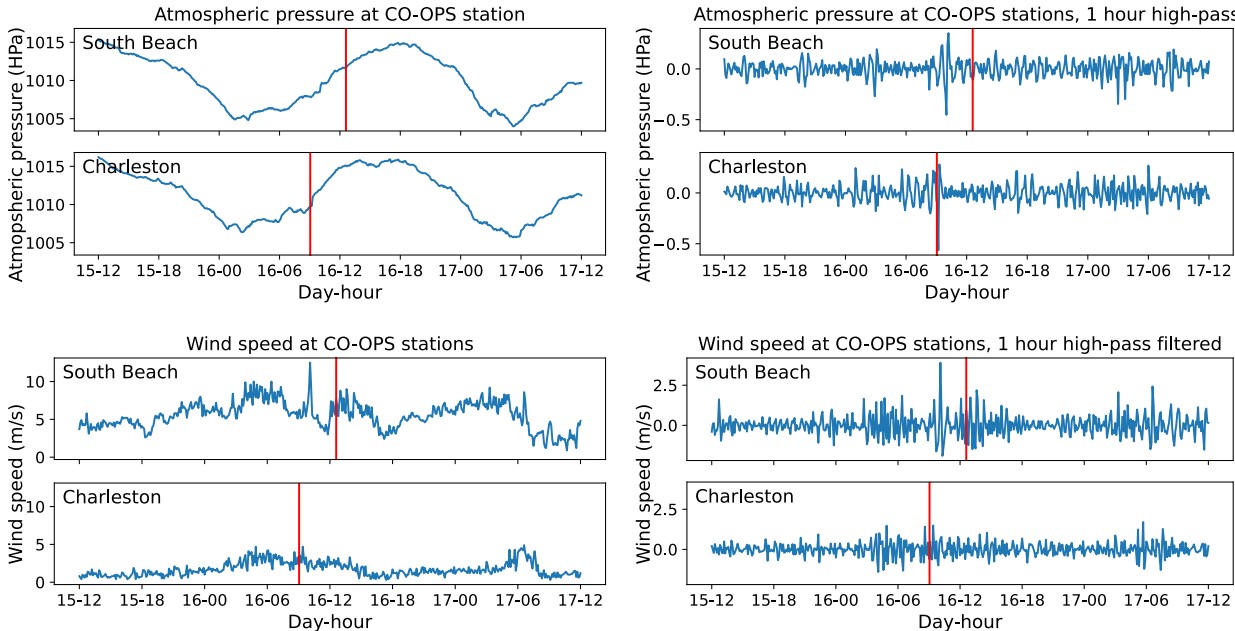

**Figure 7.** Atmospheric pressure and wind speed (left, top and bottom respectively) at South Beach and Charleston tide gages from January 15, 12:00 – January 17, 12:00, 2016. Red vertical lines indicate January 16, 12:36 and 9:03, for South Beach and Charleston respectively, i.e. the approximate onset of unusual water level fluctuations at the tide gages. (Right) one-hour high-passed version of the left.

that recorded the largest water level fluctuations - La Push and Charleston - both show considerably greater energy under the ~5 min peaks (taking account of the log-scale). For example, at La Push, the energy under the ~5 min peak is about 3 times that under the ~17 min peak. In addition, large waves were detected off the shelf (i.e. at the DART bottom sensors) hours after the initial onset of large runup events, indicating that the returning waves were able to travel past the shelf into deeper water. As such, while shelf resonance may have enhanced the runup events on January 16, they are not likely to be the primary driver.

Extreme runup can also be caused by a phenomenon known as a meteotsunami. In this mechanism, a large atmospheric disturbance travels at the shallow water speed and creates a tsunami-like runup (Monserrat et al., 2006). As described earlier, the largest atmospheric pressure anomaly from January 15 12:00 to January 17 12:00 is under 1 HPa. The wind speed anomaly over this period is about 5 m/s at South Beach and 2.5 m/s at Charleston. In contrast, the atmospheric pressure anomaly and wind speed that led to the meteotsunami analyzed by Olabarrieta et al. (2017) are approximately 5 HPa and 15 m/s, respectively. Multiple meteotsunamis in the study of Monserrat et al. (2006) are also associated with atmospheric pressure anomaly of around 5 HPa. More recently, Anarde et al. (2020) showed that meteotusnamis can be generated by ~1-3 HPa of atmospheric disturbance. However, the periods of meteotsunamis in their study were around 20 min, which is considerably longer than the ~5 min period of the extreme runup events in this study. In addition, the meteotsunamis in Anarde et al. (2020) are associated with sustained atmospheric pressure anomaly over 80 hours of around 2 HPa (3 standard deviations). In contrast,

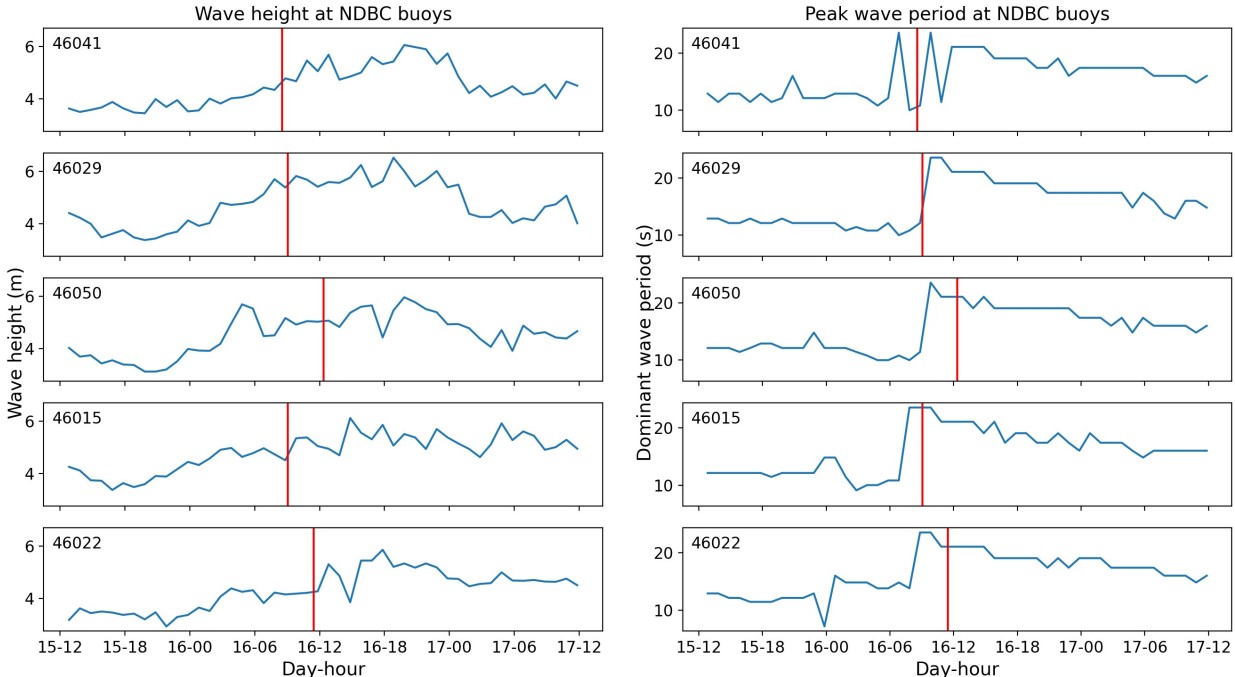

**Figure 8.** Significant wave height (left) and peak wave period (right) at NDBC buoys (ordered north to south from top to bottom) from January 15, 12:00 to January 17, 12:00, 2016. Data are recorded at 1-hour intervals. Red vertical lines represent January 16, 8:33, 9:03, 12:21, 9:03, and 11:27 from top to bottom, i.e. the approximate onset of unusual water level fluctuations at the nearby tide gages.

the sustained atmospheric anomaly from around the extreme runup events in this work (from January 16, 9:00 to January 17, 6:00) was about 0.28 Hpa. The meteotsunami events described by Olabarrieta et al. (2017) and Sheremet et al. (2016) involve
one primary large wave (soliton) sometimes followed by large waves with rapidly decaying amplitudes (over a few minutes and with periods of incident waves). More recently, Shi et al. (2020) showed that long-lasting wave trains of meteotusnami waves can be generated. In comparison, the January 16 extreme runup events had periods of ∼5 min, while the meteotsunami wave trains in Shi et al. (2020) had periods of hours. As such, while possible, January 16 runup events are not likely meteotsunamis due to the lack of strong atmospheric pressure and wind speed anomaly, and the considerably different amplitude-decay and
period characteristics from what is discussed in the meteotsunami literature.

A third possible generation mechanism for extreme runup events is considered here. As described in the previous section, one of the most striking features of the environmental conditions leading to and during the occurence of the large runup events is the rapid and significant increase in wave energy in low (<0.06 Hz) frequency swells. This observation and the observation of a ∼5-minute period in water level response at the tide gages suggest a connection between the extreme runup events and
infragravity waves. Specifically, it is known that infragravity waves have periods corresponding to those of wave groups, and a 5-minute period is a plausible period for wave groups when carrier waves have periods of approximately 25 s. For example,

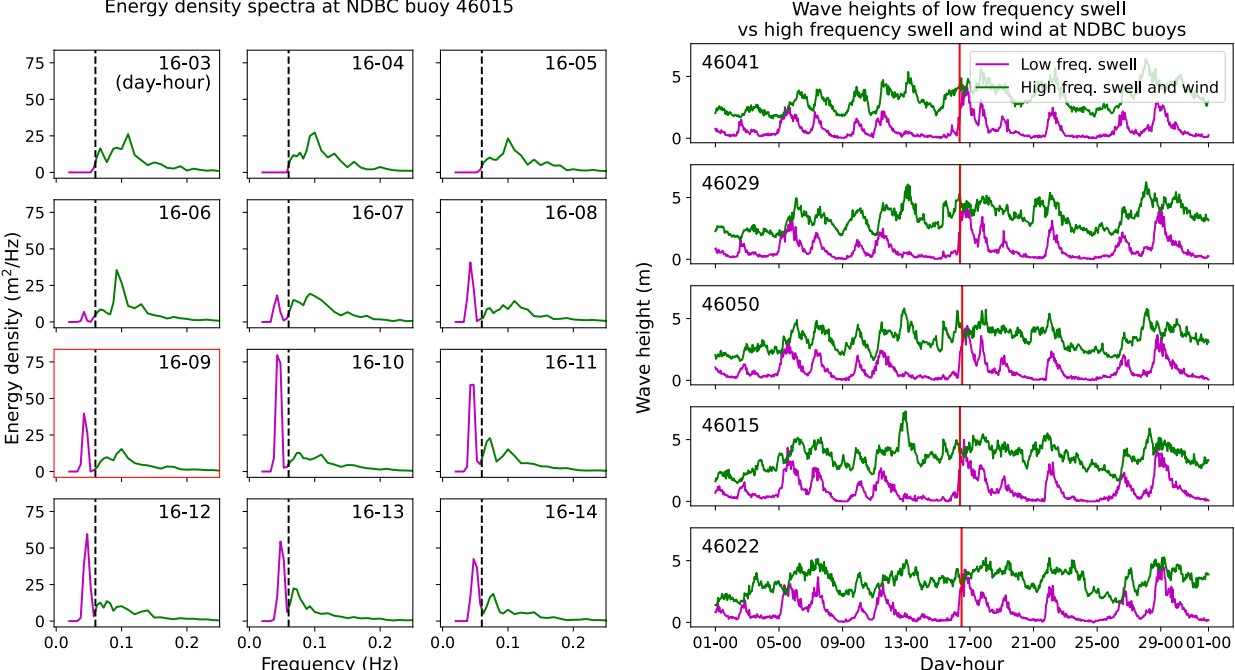

**Figure 9.** Ocean wave spectra (left) at NDBC buoy 46015 from January 16, 2016, 3:00 to 14:00. Vertical dash lines are located at a frequency of 0.06 Hz and is used to separate the swell component from the wind. Red box represents January 16, 9:00, i.e. the approximate onset of unusual water level fluctuations at the nearby tide gage. (right) Time series of significant wave height from swell and wind components calculated from the energy density spectra on January 16. Red vertical line represents January 16, 8:33, 9:03, 12:21, 9:03, and 11:27 from top to bottom, i.e. the approximate onset of unusual water level fluctuations at the nearby tide gages..

12 waves at 25 s period would make a 5-minute wave group. To verify this connection, we performed a wave group analysis on the offshore wave spectra. We use Kimura's method (Kimura, 1989; Battjes and van Vledder, 1984) - one of the most well accepted methods (Rodriguez et al., 2000) - of calculating wave group periods. In any wave group analysis method, a critical wave height is required to determine the beginning and end of each wave group. We use the commonly chosen significant wave height (Masson and Chandler, 1993; Battjes and van Vledder, 1984; Rodriguez et al., 2000) for this parameter. We find that the mean group periods of the low frequency swell (see Figure 9) are 392 s, 364 s, 378 s, 345 s, and 355 s for stations 46041, 46029, 46050, 46015, and 46022 from January 16 12:00 to 17:00, i.e. the period during which extreme runup events were observed. This implies a mean value of 367 s across all stations. This compares reasonably well with the ~5 minute (~300 s) peaks at the tide gages. If the RMS wave height, instead of the significant wave height, is used as the critical wave height, the mean group period is 188 s. This is somewhat lower than the ~300 s peaks at the tide gages, but closer to the period of fluctuations at the deep water sensors (discussed next). It is known that bound infragravity waves associated with wave groups can experience considerable shoaling under appropriate conditions (Longuet-Higgins and Stewart, 1964; Battjes et al., 2004).

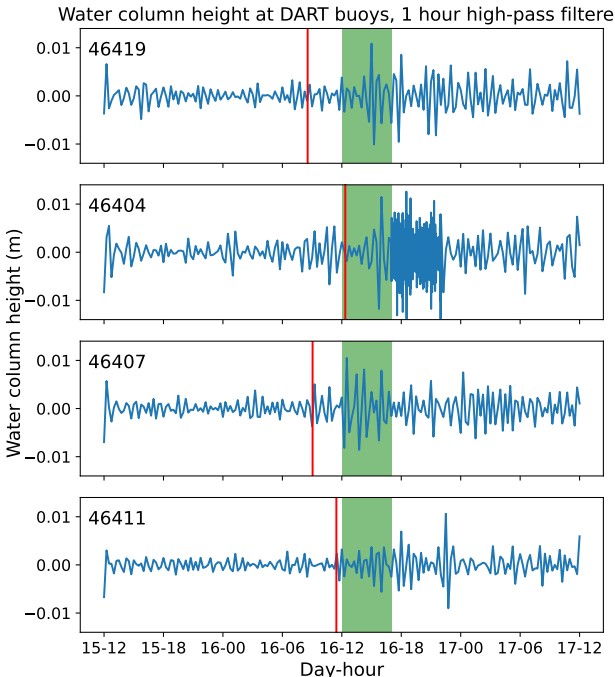

**Figure 10.** Water column height deviations (1-hour high-pass filtered) at DART sensors from January 15, 12:00 – January 17, 12:00, 2016. Red vertical lines represent, January 16, 8:33, 12:21, 9:03, and 11.27, respectively from top to bottom, i.e. the approximate onset of unusual water level fluctuations at the tide gages. Green bars indicate time period during which videos of large runup and injury reports took place, i.e. between January 16 12:00 to January 16 17:00. Between approximately January 16 17:00 to January 16 22:00, the 46404 station was in higher sampling mode.

Further detail on this mechanism, as well as a method using this mechanism to forecast similar events, are in the discussion section.

Large infragravity waves as a driver for extreme runup events is also supported by the observations of water column heights at the far offshore DART sensors. Specifically, the fluctuations of water column heights at the DART sensors started several hours after the approximate onset of unusual water level fluctuations at the tide gages. This suggests that the heights of the incoming infragravity waves were rather small compared to those of the reflected infragravity waves. This is also consistent with the findings that the most energetic infragravity waves in the deep ocean originate from the nearshore (Smit et al., 2018). To compare the periods of fluctuations at the DART sensors to those of fluctuations at the tide gages, we performed a spectral analysis on the DART recordings. We find that the peak period of the DART station 46404 during the high sampling mode is about 225 s (3 min 45 s). This is somewhat lower than the $\sim$300 s (5 min) peak at the tide gages. However, there are still considerable amounts of energy at 225 s at tide gages close to 46404, including La Push, Westport, and Garibaldi. As mentioned above, if RMS wave height, instead of significant wave height, is used as the critical wave height to determine wave groups,

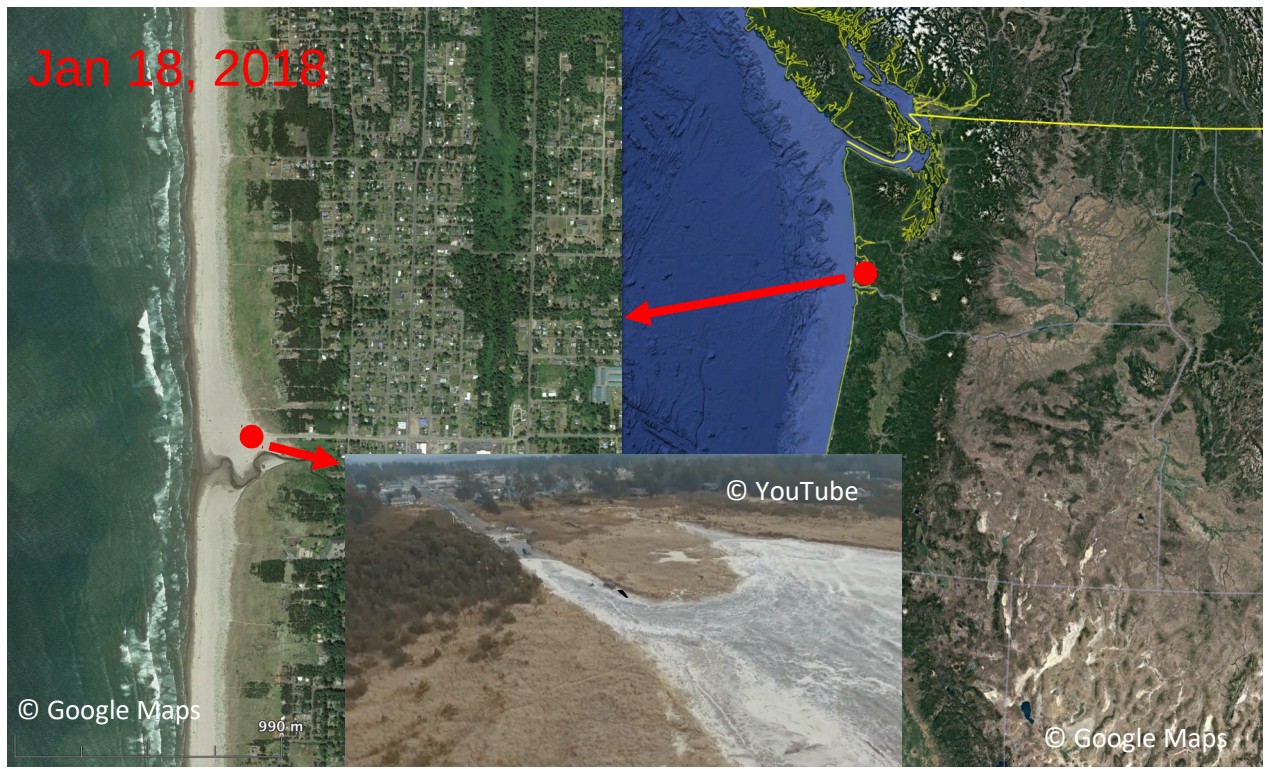

**Figure 11.** Location and a photograph from an extreme runup event on January 18, 2018 (satellite images from ©Google Maps, insert from YouTube, 2020f).

we find that the mean group period during the extreme runup events is 188 s (3 min 6 s), i.e. closer to the peak at the DART sensors. To get a sense of travel time, we assume a simplified shelf geometry and that the infragravity waves have a period of 225 s. It is then estimated that these waves would take approximately 1.1 to 1.3 hours to travel from shore to the DART sensors. The 1.3 hours value results from an assumption of a simplified shelf geometry comprised of 3 uniformly sloping segments of 210 km, 100 km, and 50 km in horizontal length, starting at depths of 3738 m, 2293 m, and 181 m, respectively. These are based on the location and depths of NDBC stations 46404 (West Astoria), 46089 (Tillamook), and 46248 (Astoria Canyon), respectively. The 1.1 hours value results from an assumption of a single uniform sloping bottom of 360 km in horizontal length, starting at the depth of 3738 m (West Astoria).

## 6   Discussion

### 6.1   Generation of waves with very long periods

The large runup events of January 16, 2016 were associated with a rapid increase of peak period and energy at low frequencies (i.e. Figure 8 and Figure 9). It is thus worth discussing how incident waves of very long periods can be generated. It is known

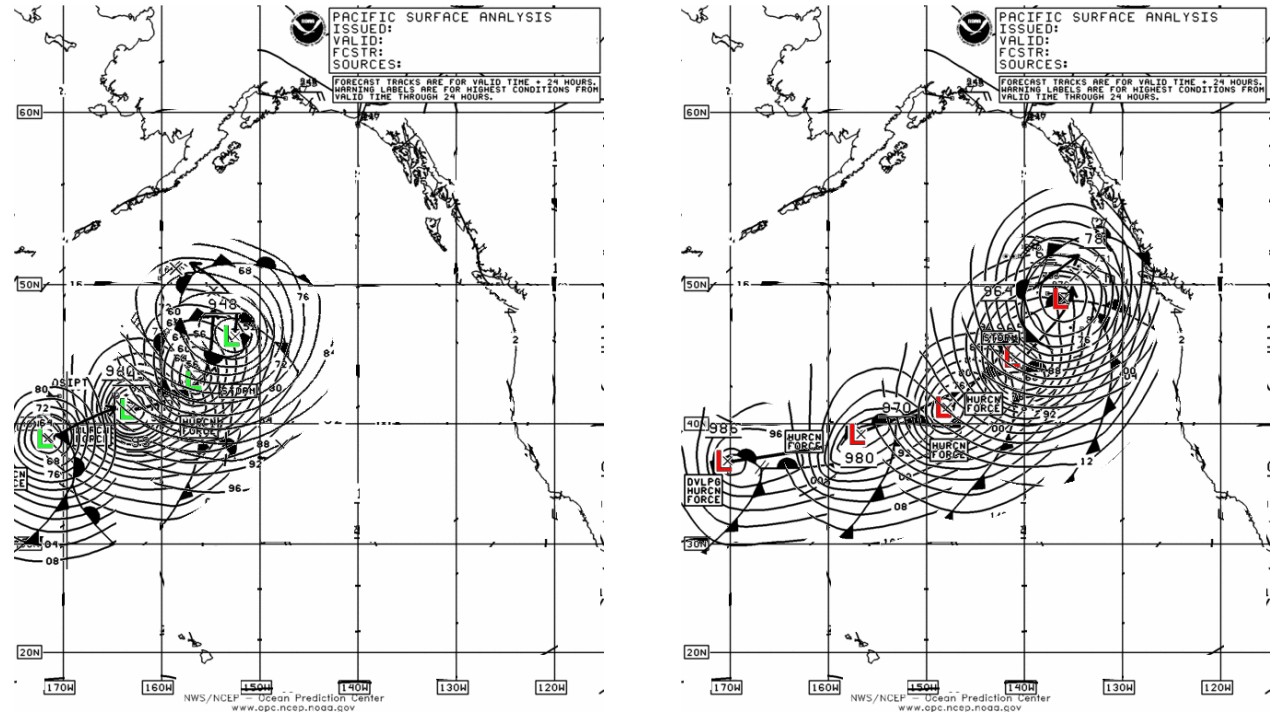

**Figure 12.** Storm tracks for January 16, 2016 (left) and January 18, 2018 (right). L represents center of storm (Elson, 2018).

that the height and period of a deep-water wave increase as the wave stays within the wind system, or fetch (Wilson, 1955; Bowyer and MacAfee, 2005). If the fetch moves in the same direction as the waves, the waves would remain in the fetch for a longer period than if the fetch was stationary, and thus grow to be higher and longer. If, in addition to moving in the wave direction, the fetch also moves at speeds very close to those of the wave groups, waves of even larger heights and periods can be generated due to the longer duration in which the waves stay within the fetch. This has been referred to in the literature as trapped fetch, dynamic fetch, effective fetch, fetch enhancement, and group velocity quasi-resonance (Dysthe and Harbitz, 1987; Bowyer and MacAfee, 2005).

The likelihood of trapped fetch contributing to the January 16 runup events can be analyzed using storm tracks and wave periods. Figure 12 (left) shows the storm track on January 16, 2016. Wave periods reported by the offshore NDBC buoys range from 19 s to 23.5 s between 2016 January 16 13:00 and 2016 January 16 17:00, during which time videos of large runup and injury reports took place (Figure 6). Using the deep-water dispersion relationship, the resulting wave group speeds were approximately 53 km/hr to 66 km/hr. By measuring distances between centers of the storms at different times it is estimated that the fetch regions were traveling at 57 km/hr. Furthermore, peak wave directions were about 265 deg clockwise from north, and are in reasonable alignment with the tracks of the storms. Therefore, trapped fetch was likely to be at least partially responsible for the January 2016 events.

As large as the January 16, 2016 runup events were, they were not the only occurence of extreme runup in this region in recent years. On January 18, 2018, video footage (YouTube, 2020f) was taken at a coast of the PNW (Figure 11) that shows recurring extreme runup events with similar characteristics on this day compared to January 16, 2016. Figure 12 (right) shows the storm track on January 18, 2018. Similar large fluctuations were also observed at the tide gages, NDBC buoys, and DART bottom sensors on January 18, 2018 (not shown). From 2018 January 18 8:00 to 2018 January 18 17:00, i.e. approximate time of sunrise to sunset on January 18, 2018, the wave periods ranged between 16 s to 19 s. The corresponding wave group speeds are approximately 45 km/hr to 53 km/hr. Analysis of the storm track shows that the storm was traveling at 53 km/hr. Peak wave directions were about 238 deg from north, which is also in reasonable alignment with the storm track. Thus, trapped fetch was likely to be in effect for the January 2018 events as well.

While it is plausible that a very large and very strong storm could generate waves of very long periods without having a trapped fetch, the trapped fetch provides a mechanism from which lesser storms could potentially generate waves of long enough periods to lead to extreme runup.

## 6.2 Generation of large infragravity waves

As shown in Figure 6, a common peak period at approximately 5 minutes is seen in the energy density spectra of the large water level fluctuations at all 6 tide gages along the PNW on January 16, 2016. In addition, a period of approximately 6 minutes can be deduced from one of the videos taken on that day (YouTube, 2020a). A period of approximately 5-6 minutes is also a plausible time-scale for wave groups.

Wave motions with periods in the time-scales of wave groups have long been known to exist (e.g. Munk, 1949; Tucker, 1950). Two mechanisms are known to generate such waves. In the first mechanism, variations of amplitudes within a wave group lead to transfer of momentum in such a way that produces a depression of the mean water level at the location of waves with greater amplitudes and an elevation of mean water level at the location of waves with smaller amplitudes. This so-called 'bound infragravity wave' travels at the speed of wave groups towards shore, and is released as the carrier waves break (Longuet-Higgins and Stewart, 1964). In the second mechanism, changes in the cross-shore location of wave breaking releases a free wave towards the shore (Symonds et al., 1982). The relative importance of the second mechanism is known to decrease with decreasing beach slope (List, 1992; Battjes et al., 2004). This suggests that in the case of relatively low beach slope (as is in this study), the first mechanism (bound infragravity waves) is most likely more important.

An important consideration of bound infragravity waves is that they shoal differently from free waves. The amplitude of a free wave shoals at a maximum that is proportional to $h^{-1/4}$, i.e. Green's Law, in shallow water. The amplitude of a bound infragravity wave can reach a maximum shoaling of $\propto h^{-5/2}$, which is a power of 10 greater than the maximum shoaling of free waves (Longuet-Higgins and Stewart, 1962). On natural beaches, however, the shoaling of bound infragravity waves tend to be less than $\propto h^{-5/2}$, due to the limited time available to reach dynamical equilibrium while the waves are traveling over a sloping bottom (Longuet-Higgins and Stewart, 1962; Battjes et al., 2004). Nonetheless, given the right conditions, bound infragravity waves can shoal to considerable heights, considering that their heights are quite small in deep water.

To illustrate the shoaling of bound infragravity waves, an analysis using a simple mathematical model is performed. The results are shown in Figure 13. Here, the amplitude profiles of the infragravity waves and carrier waves, and energy dissipation due to breaking are shown for carrier wave periods $T = 10$ s and $T = 25$ s, which respectively corresponds to conditions just before and during the January 16 events.

The model contains four regions for the infragravity waves: offshore, bound wave shoaling, free wave shoaling, and breaking. The offshore region is defined as being outside of bound wave shoaling region, i.e. $kh > 1.1$, where $k$ is the wave number and $h$ is the water depth (van Dongeren et al., 2007). In this region, the amplitudes of the infragravity waves are found from the following formulation for the water surface elevations of bound infragravity waves, valid for wave groups that are long compared to the water depth, from Longuet-Higgins and Stewart (1962):

$$320 \quad \eta_{ig} = -\frac{1}{2} \frac{ga^2}{gh - c_g^2} \left( \frac{2c_g}{c} - \frac{1}{2} \right), \tag{2}$$

where $\eta_{ig}$ is the water surface elevation of the infragravity wave, $g$ is the acceleration of gravity, $a$ is the carrier wave amplitude, $h$ is the water depth, $c_g$ is the group celerity, and $c$ is the carrier wave celerity. $a$ is calculated from linear wave theory:

$$a = a_0 \sqrt{\frac{c_0}{2c_g}}, \tag{3}$$

where $a_0$ is the offshore carrier wave amplitude, taken to be 2.5 m based on conditions of the January 16 events (Figure 8); 325   $k$ is the wave number, computed using the dispersion relationship: $\sigma^2 = gk \tanh kh$, where $\sigma$ is the radial frequency of the carrier waves; and $c_0$ is the offshore carrier wave celerity, computed with the value of $k$ at deepwater, i.e. $\tanh kh = 1$. $c$, and $c_g$ are also computed from linear wave theory, i.e. $c = \sqrt{g/k \tanh kh}$, and $c_g = c/2(1 + 2kh/\sinh 2kh)$. Infragravity wave amplitudes are computed from $\eta_{ig}$ in (2) by assuming that carrier wave amplitudes in a wave group vary from 0 (at the node) to $a$ (at the antinode) - i.e. the amplitude of the infragravity wave ($a_{ig}$) at a given depth is $a_{ig} = |\eta_{ig}/2|$.

The bound wave shoaling region is defined as the region between the shoreward end of the offshore region ($kh = 1.1$) and the onset of carrier wave breaking in the nearshore, which follows $a = 0.25h_b$. In this region, the infragravity waves shoal as $a_{ig} \propto h^{-\alpha}$, where $\alpha$ is a value between -5/2 (maximum value) and -1/4 (Green's Law), and is obtained from the relationship of van Dongeren et al. (2007), which relates $\alpha$ to the wavelength-normalized bed slope, $\beta$, defined as (Battjes et al., 2004):

$$\beta = \frac{h_x}{\omega} \sqrt{\frac{g}{h_b}}, \tag{4}$$

where $h_x$ is a characteristic bed slope, taken to be 0.01, a reasonable value of nearshore slope in this region (Cohn et al., 2019); $\omega$ is the radial frequency of the infragravity wave, obtained by assuming 12 carrier waves in a wave group (see section 5.2); and $h_b$ is a characteristic water depth, here taken as the breaking depth of the carrier waves, following van Dongeren et al. (2007). Here, 12 carrier waves per group is used with both 10 s and 25 s carrier wave periods to isolate the effects of carrier wave periods on infragravity wave shoaling. The combined effects of carrier wave period and number of carrier waves 340   on infragravity wave shoaling is not analyzed here. The relationship between $\alpha$ and $\beta$, although not explicitly stated in van Dongeren et al. (2007), is approximated from their results as $\alpha = -1.5\beta + 2$.

The free wave shoaling region is defined as the region between the onset of carrier wave breaking and the onset of infragravity wave breaking, which follows $a_{ig} = 0.25h_b$. In this region, the infragravity wave shoals as $a_{ig} \propto h^{-1/4}$, i.e. Green's Law.

The breaking region is defined as the region between the onset of infragravity wave breaking and the still-water shoreline. In this region, the infragravity wave height is computed using the long wave energy equation, as was also done by van Dongeren et al. (2007):

$$\frac{d}{dx}\left(\sqrt{gh}\frac{1}{8}\rho g H_{rms,ig}^2\right) = -D \tag{5}$$

where $x$ is the cross-shore position, $\rho$ is the water density, $H_{rms,ig}^2$ is the RMS wave height of the infragravity wave, and $D$ is the energy dissipation due to breaking, which follows as (Battjes and Janssen, 1978):

$$D = f\rho g \frac{H_{rms,ig}^2}{4} \tag{6}$$

where $f$ is the infragravity wave frequency. Furthermore, the wave reflection coefficient ($R$) is calculated at the shoreline from the relationship of van Dongeren et al. (2007) between $R$ and another parameter $\beta_H$ (Battjes et al., 2004; van Dongeren et al., 2007):

$$\beta_H = \frac{h_x}{\omega}\sqrt{\frac{g}{H_{ig}}} \tag{7}$$

where $H_{ig}$ is the infragravity wave height near the shoreline (here taken as the significant infragravity wave height, i.e. $H_{ig} = 1.416H_{rms,ig} = 2a_{ig}$, at the shoreline, i.e. $h = 0$ m). $\beta_H$ is similar to $\beta$ in (4), except for the replacement of $h$ with $H_{ig}$. van Dongeren et al. (2007) find that at low values of $\beta_H$, the reflection coefficient of the infragravity waves ($R$) is close to 0, indicating near complete energy dissipation; and at high values of $\beta_H$, the $R$ is close to 1, indicating near absence of energy dissipation. The relationship between $R$ and $\beta_H$ is approximated from the results of van Dongeren et al. (2007) as $R = 0.5\beta_H$.

Prior to discussing the results, some limitations of the simple model should be noted. Being analytical in nature, the model uses representative wave parameters, i.e. carrier waves are either 10 s or 25 s (depending on each case), and the number of waves in each group is fixed at 12. Additionally, modulation of the wave height in wave groups is assumed to be between 0 m to the full wave height ($2a$) of the carrier waves. Nevertheless, this simple model provide insights to why a change (albeit a large one) in carrier wave period leads to such a difference in infragravity wave height.

The resulting infragravity wave amplitude and energy dissipation profiles are shown in (Figure 13). The bound infragravity wave shoaling associated with 25 s carrier waves ($T = 25$ s) begins at water depth $h = 137$ m, which sharply contrasts with $h = 21.8$ m associated with $T = 10$ s. This results in the infragravity wave amplitudes ($a_{ig}$) consistently higher for $T = 25$ s than they are for $T = 10$ s at all $h$ (Figure 13a). Close to shore, however, this difference is reduced as the shoaling exponent $\alpha = 1.39$ for $T = 25$ s is less than $\alpha = 1.71$ for $T = 10$ s. At its maximum value, i.e. at infragravity wave breakpoint, $a_{ig} = 0.62$ m and $a_{ig} = 0.57$ for $T = 25$ s and $T = 10$ s, respectively (Figure 13c).

The difference in $a_{ig}$ increases considerably, however, in the infragravity wave breaking region, as the energy dissipation ($D$) is significantly smaller for $T = 25$ s than it is for $T = 10$ s (Figure 13d). At its maximum, $D$ for $T = 25$ s is about half of that for $T = 10$ s. This results in $a_{ig} = 0.44$ m and $a_{ig} = 0.26$ m for $T = 25$ s and $T = 10$ s, respectively, at the shoreline

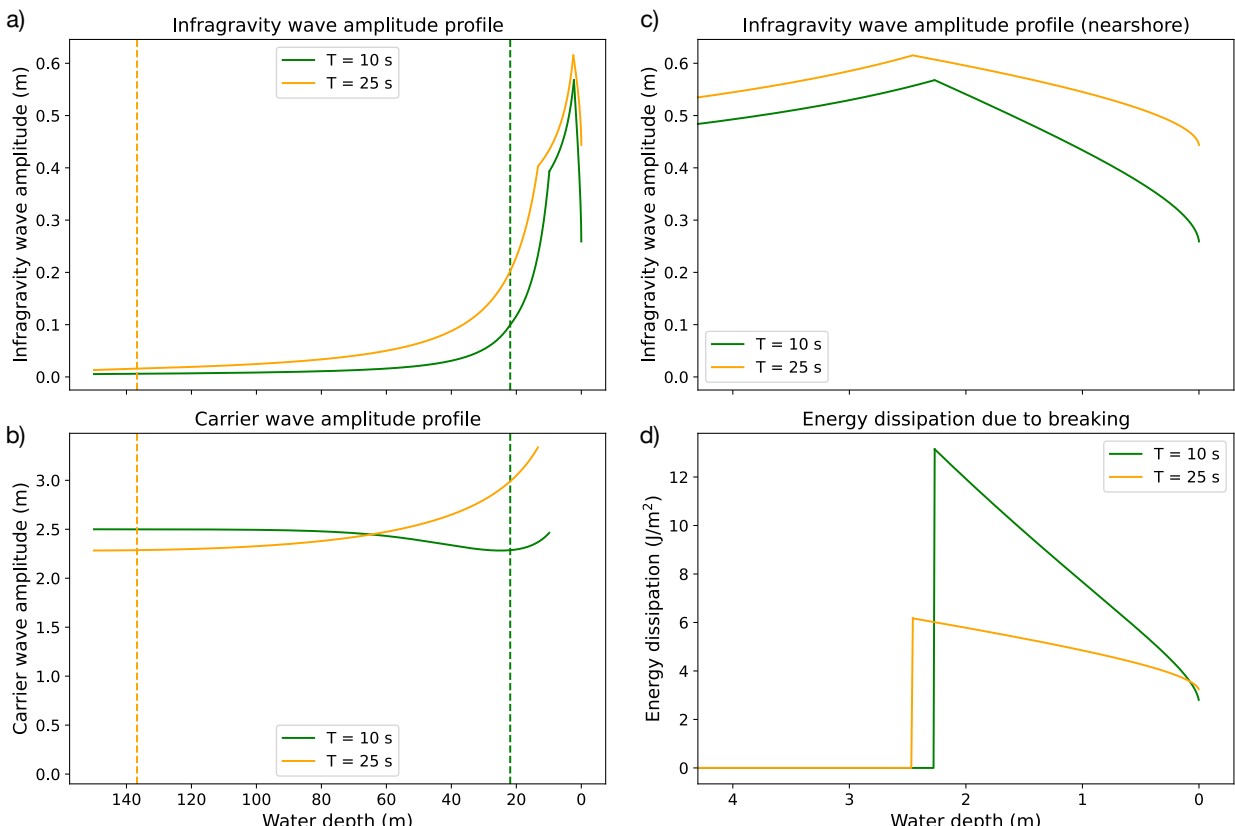

**Figure 13.** Infragravity (a, c) and carrier (b) wave amplitude profiles and energy dissipation rate (d) for 10 s (green) and 25 s (orange) carrier waves. Dashed lines indicate the locations where infragravity waves first enter the shoaling region. Infragravity amplitudes are computed with (2) seaward of the shoaling region, as $\propto h^{-\alpha}$ in the shoaling region up to the breakpoint of the carrier waves, as $\propto h^{-1/4}$ from the carrier wave breakpoint to the infragravity wave breakpoint, and as (5) shoreward of the infragravity wave breakpoint.

($h = 0$ m). The differences in the two cases are accounted for in their reflection coefficients ($R$), which are $R = 0.79$ and $R = 0.42$ for $T = 25$ s and $T = 10$ s, respectively. $R$ is important because it determines the fraction of the wave height at the shore ($H_{ig} = 2a_{ig}$ at $h = 0$ m) that is able to be reflected back, which corresponds to standing wave oscillations associated with wave runup (Miche, 1951; Guza and Bowen, 1976; Guza et al., 1984), and especially wave runup in the infragravity frequencies (Guza et al., 1984). $H_{ig}$ and $R$ together imply that the reflected infragravity wave heights are $RH_{ig} = 0.70$ m and $RH_{ig} = 0.22$ m for $T = 25$ s and $T = 10$ s, respectively, which represents a ratio of 3.2:1.

The plausibility of maintaining a high infragravity wave growth rate and low energy dissipation is also supported by observations at the DART sensors (Figure 10). It is seen that the large fluctuations of water column height occur hours after the first occurrence of large water level fluctuations at the shore. This suggests that the incoming infragravity waves, before shoaling are not large enough to cause a significant response at the sensors, whereas the outgoing infragravity waves are able

to produce a significant response due to having achieved considerable shoaling and low dissipation before being reflected away from shore. And as stated earlier, this is also consistent with the finding that the most energetic infragravity waves in the deep ocean originate from the nearshore (Smit et al., 2018).

## 6.3 A predictor for extreme runup events due to large infragravity waves

As shown in the results section, an important observation of the January 16 large runup events is that these events are connected with rapid growth of wave energy in low frequency swells. This connection can be exploited to explore a predictive tool for similar large runup events. The goal of this predictor is to use a metric of ocean waves to predict a metric of water levels at the shore. One approach to this is to use a cutoff frequency to identify the low frequency component of the ocean wave spectra. However, it was found that the subjectivity of the cutoff frequency makes the predictor less robust. Instead, we explored an approach that uses the negative moments of the ocean wave spectra. In this approach, negative moments (1) at NDBC buoys were correlated to representative measures of the intensity of water level fluctuations at the tide gages. NDBC and tide gage pairs were determined based on proximity. Square roots of various negative moments are nondimensionalized by their mean, e.g. $\sqrt{m_{-1}}/\overline{\sqrt{m_{-1}}}$. Water levels RMS - hereinafter referred to as $\eta_{\mathrm{rms}}$ - was chosen to represent the magnitude of the water level fluctuations. It is also normalized with its mean, i.e. $\eta_{\mathrm{rms}}/\overline{\eta_{\mathrm{rms}}}$.

We tested several nondimensionalized negative moments including $\sqrt{m_n}/\overline{\sqrt{m_n}}$, for $n$ from 0 to -6. We find that the fit is improved as $n$ becomes more negative, starting from 0, reaching optimal fit at $n$ of -4 or -5, and then becomes worse from $n = -6$. For example, the $R^2$ of fit for 46041-La Push station pair are 0.55, 0.655, 0.733, 0.788, 0.820, 0.833, and 0.830, respectively for $n$ decreasing from 0 to -6. It was deemed that $\sqrt{m_{-4}}/\overline{\sqrt{m_{-4}}}$ produced the best results considering the fit for all 5 station pairs. Figure 14 shows a comparison of $\sqrt{m_{-4}}/\overline{\sqrt{m_{-4}}}$ with $\eta_{\mathrm{rms}}/\overline{\eta_{\mathrm{rms}}}$ for several months in 2016. Figure 15 shows the resulting relationships for this choice of negative moment for years 2016-2018. The fit appears reasonable for 4 out of the 5 station pairs, with $R^2$ ranging from 0.726 to 0.821, excluding the 46022 - Crescent City pair. Crescent City's harbor, which is where the tide gage is located, is known to be susceptible to wave resonance from the shelf (Allan et al. 2012, Lu et al. 2014 and Figure 6). The slopes of the fit are close to unity, and y-intercepts are close to 0 for all station pairs except Crescent City. This suggest that the data from these 4 station pairs could be collapsed into a single relationship (not done here). Furthermore, when data points from the extreme runup events of January 16, 2018 and January 18, 2018 are overlayed, it is clear that both $\sqrt{m_{-4}}/\overline{\sqrt{m_{-4}}}$ and $\eta_{\mathrm{rms}}/\overline{\eta_{\mathrm{rms}}}$ during these events are on the far end of their respective range, lending confidence to the predictive abilities of these relationships. We also present an alternative relationship, in Figure 16, with a perhaps more physically based nondimensionalization. Here, $\sqrt{m_{-4}}$ normalized by $f_{m2}^2/\sqrt{m_0}$, where $f_{m2} = \sqrt{m_2/m_0}$ is the zero upcrossing wave frequency. The fit is a power law and is somewhat worse than the fit from the $\sqrt{m_{-4}}/\overline{\sqrt{m_{-4}}}$ normalization.

The relationship described here is not only useful in predicting future extreme runup events, but can also be helpful in understanding the frequency of occurrence of these events. Figure 17 shows the monthly means of $\sqrt{m_{-4}}/\overline{\sqrt{m_{-4}}}$ and $\eta_{\mathrm{rms}}/\overline{\eta_{\mathrm{rms}}}$ for years 2016-2018 at each site. It it seen that for four out of the five sites (except Crescent City), the monthly means of both $\sqrt{m_{-4}}/\overline{\sqrt{m_{-4}}}$ and $\eta_{\mathrm{rms}}/\overline{\eta_{\mathrm{rms}}}$ during the winter months can be as high as more than triple of those during the summer months.

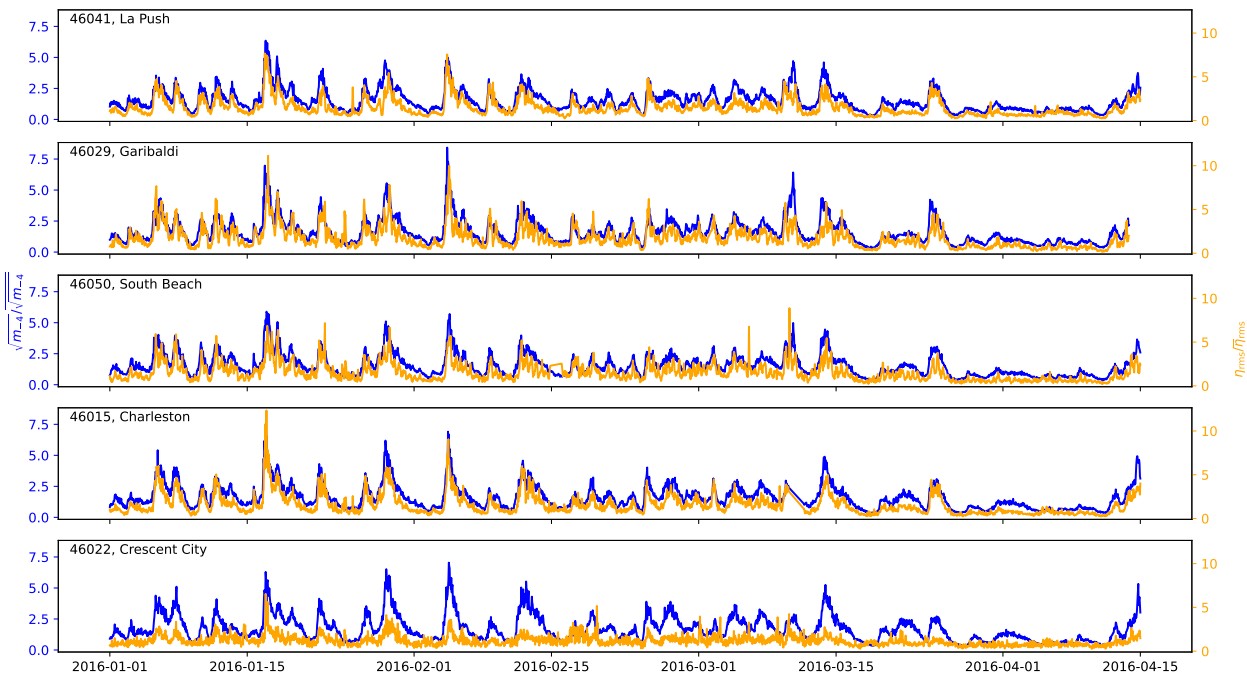

**Figure 14.** $\sqrt{m_{-4}}/\overline{\sqrt{m_{-4}}}$ (blue) of the offshore ocean wave spectra and $\eta_{\mathrm{rms}}/\overline{\eta}_{\mathrm{rms}}$ (orange) at the onshore tide gages, computed using a 0.5-hour window, from 2016 January 1 to April 15.

This suggest that the type of extreme runup events discussed in this work occur much more frequently during winter months than they occur during summer months. One reason why these extreme large runup events tend to occur more frequently during the winter months than they do in the summer months is perhaps related to the fact that the wave periods in this region is much greater in the winter months than they are in the summer months. As we have shown in the previous section on generation of large infragravity waves, long wave groups facilitate the reduction of energy dissipation.

## 7   Conclusions

This work presents an analysis of observations of unusually large runup events that occurred along the PNW coast on January 16, 2016. On this day, video recordings and injury reports document multiple extreme runup events – with horizontal excursions exceeding a hundred meters and periods of minutes – occurring along approximately 1000 km of coastline within 5 hours of each other. Environmental conditions leading up to and during the large runup events are presented.

The observations show that the large runup events are strongly associated with a rapid increase in wave energy at low frequencies, i.e. the arrival of incident waves with very long periods. In addition, water level measurements at the tide gages show a ~5 min peak period during this time. The arrival of incident waves with very long periods can be explained by the

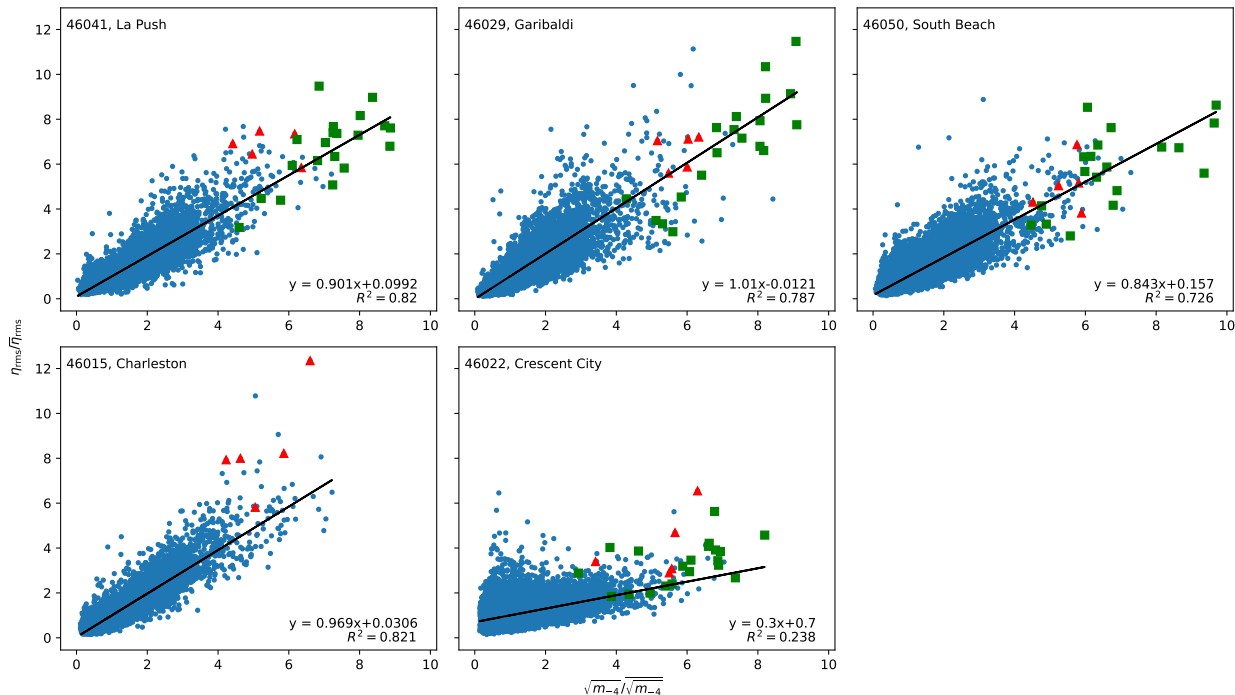

**Figure 15.** Onshore $\eta_{\mathrm{rms}}/\overline{\eta_{\mathrm{rms}}}$ at the tide gages computed from a 0.5-hour window versus $\sqrt{m_{-4}}/\sqrt{\overline{m_{-4}}}$ from the offshore ocean wave spectra for year 2016 to 2018. Red dots correspond to the duration between 2016 January 16 13:00 to 2016 January 16 17:00, i.e. time during which video recordings of large runup and injury reports took place. Green dots represent the duration between and 2018 January 18 8:00 to 2018 January 18 17:00, which represents a period during another series of extreme runup events that has been captured on video.

existence of trapped fetch, which occurs when the fetch moves in the same direction and at speeds close to those of the waves groups. Analysis of storm tracks show that trapped fetch was indeed in effect for this event leading to the large runup events. The ∼5 min peak period at the tide gages suggests a link to wave groups and infragravity waves. It is shown using a simple model that a very large carrier wave period results in a very large infragravity wave amplitude at the shore yet maintaining low energy dissipation. This explanation is supported by far offshore bottom sensors, which detected large waves hours after the first large runup events were observed onshore, suggesting that the reflected waves were larger than the incoming ones.

Using the link between low frequency wave energy and large runup events, a predictor for similar types of large runup events is developed. The predictive ability is seen to be reasonable for 4 out of the 5 tide gages used in the study, and for two different sets of large runup events. Results from this predictive method suggest that the type of large runup events discussed in this work tend to occur much more frequently in the winter months than they do in the summer months (Tillotson and Komar, 1997, and Figure 1). This is likely owing to the fact that the wave periods are much longer during the winter months (e.g. median = 12.9 s in January) than they are in the summer months (e.g. median = 8.3 s in August). This would result in longer infragravity

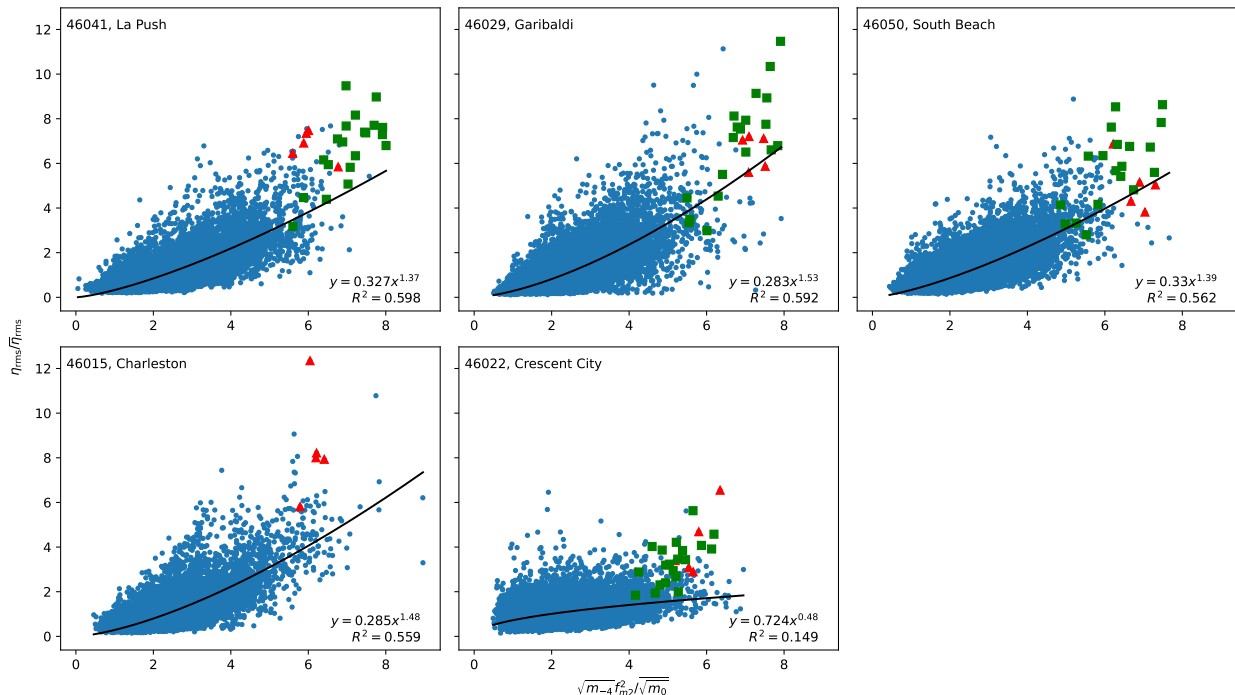

**Figure 16.** Onshore $\eta_{\mathrm{rms}}/\overline{\eta_{\mathrm{rms}}}$ at the tide gages computed from a 0.5-hour window versus $\sqrt{m_{-4}}f_{m2}^2/\sqrt{\sqrt{m_0}}$ from the offshore ocean wave spectra for year 2016 to 2018. Red dots correspond to the duration between 2016 January 16 13:00 to 2016 January 16 17:00, i.e. time during which video recordings of large runup and injury reports took place. Green dots represent the duration between and 2018 January 18 8:00 to 2018 January 18 17:00, which represents a period during another series of extreme runup events has been captured on video.

waves and larger associated runup. The performance demonstrated by this predictive method may be helpful to future efforts in developing forecasting tools for extreme runup events, with the aim of issuing warnings to the public.

A limitation of our study is the fact that measurements from tide gages were the only nearshore measurements available. As as result, we were not able to examine e.g. the cross-shore wave height profile of the infragravity waves. A future study could benefit from the deployment of multiple e.g. pressure sensors and current meters nearshore. A challenge of this would be planning the deployments in anticipation of a set of upcoming extreme events. For this purpose, the predictive method presented in this work may be useful.

*Data availability.* The data used in this work are publicly available via sources referenced.

*Video supplement.* The videos referenced in this work are available as cited.

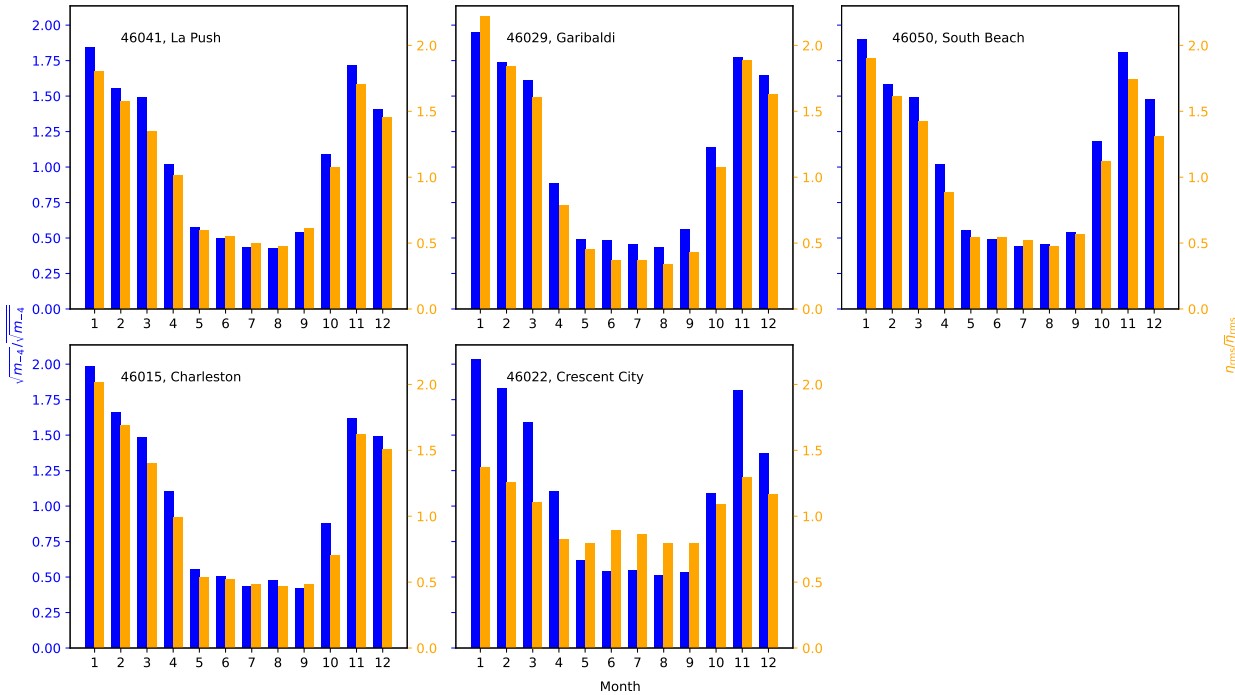

**Figure 17.** Monthly $\sqrt{m_{-4}}f_{m2}^2/\sqrt{m_0}$ (blue) from ocean wave spectra and $\eta_{\mathrm{rms}}/\overline{\eta_{\mathrm{rms}}}$ (orange) at the tide gages, computed from 0.5-hour window, for year 2016.

*Author contributions.* Özkan-Haller, Holman, and Ruggiero acquired funding for this work. Ozkan-Haller formulated the research goals and supervised the execution of the research. Holman and Ruggiero provided comments on the manuscript. Jensen, Elson, and Schneider contributed to the provision of data used in this research and provided research ideas. García-Medina contributed to research ideas and facilitated the investigation. Li carried out the investigation and prepared the manuscript with contributions from all co-authors.

*Competing interests.* The authors declare that they have no conflict of interest.

*Acknowledgements.* This work was funded by the National Science Foundation under award OCE-1459049. The authors also thank Jeremiah Pyle, Tyree Wilde, Sven Nelaimischkies, Brian Nieuwenhuis, Troy Nicolini, David Bright, John Lovegrove, and others at the National Weather Service for their collaboration. Thanks are also given to Peter Nielsen and Andrew Kennedy for their helpful comments and suggestions.

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
