# Peer review of "Observations of extreme wave runup events on the U.S. Pacific Northwest coast"

_Natural Hazards and Earth System Sciences, 2020_

## Author Comment (AC2)

[Figure]

Figure 1. Onshore Water level RMS using a 0.5-hour window length at the tide gages versus $\sqrt{m_{-3}f_{m2}^2}$ of the offshore ocean wave spectra for year 2016 to 2018. Red dots correspond to the duration between 2016 January 16 13:00 to 2016 January 16 17:00, i.e., time during which video recordings of large runup and injury reports took place. Green dots represent the duration between and 2018 January 18 8:00 to 2018 January 18 17:00, which represents a period during another series of extreme runup events has been captured on video.

[Figure]

Figure 2. Cross-shore amplitude profile of incident waves with 5 m wave height and 10 s (green) and 25 s (orange) wave periods. Dashed lines indicates where the incident waves reach shallow water. Dotted lines show where the incident waves start to break.

[Figure]

Figure 3. Cross-shore amplitude profile for bound infragravity waves associated with wave groups each consisted of 12 individual waves. Maximum individual wave height is 5 m. Individual wave periods are 10 s (green) and 25 s (orange). Dashed lines indicates where the incident waves reach shallow water. Dotted lines show where the incident waves start to break.

---

## Author Response (AR1)

**General comments**

This manuscript investigates the mechanisms leading to a series of very large run-up events that took place within a couple of hours at different locations along the US Pacific Northwest coast. The authors show that the series of extreme runup events coincides with the arrival of very long swells at the coast and hypothesize that these long swells were responsible for the generation of unusually large (initially bound) infragravity waves that ultimately led to the large, 5-6 min period, run-up events that were filmed by beachgoers.
The manuscript addresses a topic that is relevant to the scientific community and presents some interesting results. The analysis is however limited by the type of data that is available (e.g., no nearshore wave measurements except the 1min-resolution tide gage water level data) that prevents strong conclusions to drawn. I think that it is acceptable as long as the limitations of the analysis are clearly stated (in the core of the manuscript but also in the conclusions).

We appreciate the reviewer for their thorough review. Regarding nearshore data limitation, we added a statement in the beginning of our Methods section stating that only nearshore measurements available to us were from the tide gages along the coast, and that future studies could benefit from deployment of additional nearshore sensors. We also added a paragraph near the end of our Conclusions section stating this limitation and potential improvements and challenges in a future study (see lines 418-422 in the revised manuscript).

This is a point that needs improvement in my opinion (see comments on below the interpretation of the DART data for instance). Furthermore, several aspects of the methodology, particularly in Section 6.2, need to be clarified and possibly contain errors (see detailed comments in the next section).
I therefore recommend major revisions.

We really appreciate the reviewer's thorough review and comments. Following the reviewer's comments and suggestions, we have made major improvements to our paper, especially in the presentation of our data analysis and mathematical model. Please see our response to each specific comment below.

**Specific comments**

1.
2. The paper refers a few times to the runup of bound infragravity waves. I find the formulation confusing as the infragravity waves that run up the beach are not bound anymore (released in the surf zone).

   This is coming back at different locations within the paper, in slightly different forms. For instance it is stated line 265 that "th[e] bound wave travels at the speed of the wave groups towards the shore and as a free wave [...] after reflection" which is not strictly

correct as IG waves are not bound until the shore. Or in line 312 the authors discuss the "dissipation of the bound infragrvaity waves" and refer to van Dongeren et al. 2007 to do so, while this paper examines dissipation close to the shoreline, i.e. at a location where the IG waves are not bound anymore.

We thank the reviewer and agree on this point. In our revision, we now use 'bound infragravity waves' only to refer to incoming infragravity waves prior to incident wave breaking. We now refer to our extreme runup events as due to resonant growth of infragravity waves to avoid possible inference that the infragravity waves were still bound while runup events were occurring (see title of section 6.3). Regarding line 265, we have revised the statement to "th[e] bound wave travels at the speed of the wave groups towards the shore, is released as the carrier waves break." Regarding line 312, we have appropriately removed the word "bound," as we are indeed examining energy dissipation of free infragravity waves very close to shore (see line 337 in the revised manuscript).

3. Line 32: I don't think the reference to Roeber and Bricker is appropriate as an example in which trapped waves lead to extreme runup (they actually explicitly state that "Resonant amplification over the reef flat [...] did not contribute significantly...").

We thank the reviewer and agree with this assessment. In our revision, we now refer to this work as an example of a different cause of large runup, which is due to energetic infragravity waves generated by abrupt breaking of incident waves (see line 32-33 in revised manuscript).

4. Line 62: "the amplification of low frequency motion was found to decrease dramatically during storms". What does that mean? How does it relate to the previous statements?

By low frequency runup we meant runup associated with infragravity wave frequencies. We have made this clarify in our revision (line 63 in revised manuscript). This statement is not related to the previous statements but is a key finding from the previous work in discussion.

5. Line 106: I do not understand the sentence "Data is [...] averaged over 1 minute to produce 1 min timeseries"? Is 1 minute the sampling interval or the duration? (probably the first one?)

We agree that this did not read very clearly in the original manuscript. In the revision, we have revised the wording to "Data is sampled at 1 Hz and recorded at 1-minute intervals as 1-minute averages." See line 111 in revised manuscript.

6. Lines 180-182: I find that the time delay between observations of "larger than usual" oscillations at the DART sensors and at the tide gages difficult to visualize and as a result the numbers given in the text seem quite arbitrary (I could argue based on visual

inspection of figure 5 that the largest waves at Garibaldi occur several hours after the point identified as the approximate onset of the anomalous water level fluctuations... which would in turn change all the time lags). As this time delay seems important to the current narrative (it is used later to support the fact that the observed large oscillations at the DART sensors could be due to IG wave reflection), it would be useful to define periods with "higher than usual magnitudes" in a more systematic way (e.g., it starts when the elevation is larger than X*std(eta)?).

We appreciate this comment. We have followed the reviewer's suggestion and now use 2.5 times std(eta) as a threshold to determine the beginning of large water level fluctuations. As a result, the onset of large water level fluctuations at the tide gages are now differ by a small amount from what we had before (i.e., chosen by visual inspection). See revised Figure 5, 7, 8, 9, 10 and revised caption to Figure 5, 7, 8, 9, and 10. These changes do not alter our findings.

7. Lines 191-193: As wave direction cannot be inferred from a point measurement of surface elevation/water column height, I do not think it is fair to talk about "large offshore directed waves detected at the DART bottom sensors". It should be formulated in such a way that it is obvious that it is only a hypothesis, which, if I understand correctly, mostly relies on the fact that there is some time lag between the moment at which the largest oscillations are observed at the tidal gages and at the DART sensors. See also previous comment.

We agree with this assessment. In our revision, we have removed the words "offshore directed" in this sentence (see lines 200-201 in revised manuscript). Later in our manuscript (lines 243-244 in revised manuscript), we write that the fact that the large waves at the DART buoys were detected after the large runup events onshore "suggests that the heights of the incoming infragravity waves were rather small compared to those of the reflected infragravity waves."

8. Line 194: At several tide gages (Crescent City, South Beach, Westport) the spectra seem to peak in the 13-22min band (figure 6), and not at 5 min, which suggests that the water levels at these gages (and thus possibly the runup) is dominated by motion in this lower frequency band (associated with resonance in the text). I am not sure this is consistent with the statement that resonance is of secondary importance to explain the extreme water levels. Unless the variance contained in this lower frequency peak is actually less than the variance in the 5min peak (hard to evaluate visually because of the log-scale in the frequency axis)?

We realize that the spectra indicate that at some gages there were strong water level motions in the longer, 13-22min periods. However, video footage (https://youtu.be/JMYLvSsWR_g) shows that the extreme runup events were much closer to the 5 min period. The tide gages are also well inside the inlets. As some of the videos show, the heights of the bores associated with the extreme runup events were

considerably reduced as they traveled further into the inlets. Furthermore, the two gages that show the largest fluctuations – La Push and Charleston (we do not include Crescent City here because it is known to amplify shelf resonance from its harbor) – show a clearly stronger signal in the 5 min period (considering the log scale). We added these clarifications in our revision (see lines 197-200 in the revised manuscript).

9. Lines 195-207. There has been a number of recent publications on coastal hazards induced by meteotsunamis such as Shi et al., 2020 (Nature communications) and Anarde et al. 2020 (JGR-Oceans) that could be relevant to this discussion. For instance Shi et al., 2020 showed that both single-peak meteotsunami waves and long lasting meteotsumami wave trains could be generated (thus not only soliton like waves as mentioned in the manuscript). Anarde et al., 2020 showed that pressure disturbances much lower than the 5HPa mentioned in the manuscript could trigger significant meteotsunamis.

We thank the reviewer for making us aware of these new works on meteotsunamis. We now cite these papers in our revision and added their results into our discussion. We note that in Shi et al., the large wave trains have periods on the order of hours (compared to our 5 min signal). We also note that the pressure disturbances in this work (<1HPa) is still on the low side of the pressures found to generate meteotsunamis in the study of Anarde et al. (~1-3HPa). The runup period (~5 min) in this work is also lower than the meteotsunami period (~20 min) in Anarde et al. We have added this discussion in lines 209-218 in the revised manuscript. We also modified the last sentence of the paragraph on this discussion to state that "while possible," the events in our work are "not likely" meteotsunamis due to the differences described above.

10. Lines 211-212: The fact that the long waves that are responsible for the extreme run- up events were generated as bound infragravity waves is a key result of this paper which is in my opinion not sufficiently supported. I understand that data availability limits the analyses that can be conducted, but still feel that stating that 5-min is a "plausible period for wave groups when carrier waves are 25 s" (see also line 260) is in itself not a very convincing statement (for such a key result). Have the authors considered using the measured spectra at the wave buoys to reconstruct a timeseries using a random phase (which is probably not a too bad assumption in these depths) and examine the group structure? They could for instance look at the spectrum of the envelope of such a reconstructed timeseries and check that it indeed peaks around 5 min? That would lend some additional support to the fact that such a wave field could indeed force bound waves in the proper frequency range.

We agree with this assessment and per the reviewer's suggestion conducted a wave group period analysis. We used Kimura's (1980) method of computing the mean wave group period directly from the energy spectra (which does not require generation of time series). We note that there is an inherent uncertainty with any method of wave group period calculations, which arises from how the wave group is defined. A wave

group is usually defined as consecutive waves with wave heights above some critical wave height. We used the significant wave height as this critical wave height. The significant wave height is a common choice of critical wave height (e.g., Kimura 1980, Battjes and van Vledder 1984, and Rodriguez et al. 2000), but the resulting group period can vary if a different threshold is used. Given this, we computed a mean group period of 367 s for the low frequency swells during the period of observed large runup events. We feel that this agrees reasonably with the ~300 s peaks of the tide gage water level signals. We have added a description of this analysis and its results in our revision (see lines 227-237 in the revised manuscript).

11. Line 222: What is the typical period of the waves measured by the DART sensors during the period where they recorded at high temporal resolution? Is it indeed close to 5min and therefore consistent with the 5-min peaks in the tidal gage spectra?

   The typical period of waves measured by DART during the high temporal resolution recording was about 225 s (3 min 45 s). It is a bit lower than the ~300 s (5 min) peaks of the tide gage spectra. However, there are still considerable amounts of energy at 225 s at nearby tide gages, including La Push, Westport, and Garibaldi. In addition, if RMS wave height, instead of significant wave height, is used as the critical wave height to determine wave groups, we find that the mean group period during the extreme runup events is 188 s (3 min 6 s). We have added this analysis in the revised manuscript (see lines 246-252 in the revised manuscript).

12. Lines 276-279 (and following): van Dongeren et al. 2007 (and I expect that Battjes et al. as well, although I haven't re-checked) used shoaling zone data in off-resonant conditions to examine the relation between the growth rate (alpha) and the normalized bed slope (beta) and demonstrate that the infragravity wave amplitude was bound between $h^{-1/4}$ and $h^{-5/2}$. This means that the relation between beta and alpha (that is used in this paper) was derived for conditions in which the carrier waves were not in shallow water, contrary to what is stated at the start of this paragraph (and a few times later on, e.g., line 295).

   We thank the reviewer for the correction. We previously had indeed incorrectly used the van Dongeren et al. 2007 formulation by applying it only in shallow water (for the carrier waves). As the reviewer correctly pointed out, the formulation should be applied to the entire shoaling region for infragravity waves. We have now corrected our model by using the formulation for the entire region shoreward of kh = 1.1, which corresponds to the largest kh in van Dongeren et al. 2007. And as we state in the response to the following comment, we have rewritten all the paragraphs describing our model for greater clarity (see lines 306-352). We specifically state the region in our model where we have applied the van Dongeren et al. 2007 in lines 315-316 in the revised manuscript.

13. Lines 291-305: I find it difficult to understand what the model is exactly doing, and even more difficult to follow the description of the model outputs that follow in the second paragraph. Some additional information is needed (see questions/remarks in bullet points below). A figure showing the cross-shore evolution of short wave and infragravity wave heights could also help.

We agree with the reviewer and have rewritten all the paragraphs on the model (see lines 306-352 in revised manuscript). We now fit the descriptions of our model naturally with the descriptions of the bound infragravity wave shoaling, so that the reader can see more clearly how it is applied. We also now include all equations used in the model (see equations 2 – 4, and inline equations in the following paragraph), as well as all input parameters and their justifications. In addition, we improved our model by using a formulation for infragravity wave water levels seaward of the shoaling region (equation 3 in revised manuscript) that is valid for all water depths – as long as the group length is great compared to depth – instead of a formulation valid for deepwater. Finally, we now include a figure showing the cross-shore profile of carrier and infragravity wave amplitudes (Figure 13).

- Line 294: I would expect that the infragravity wave amplitude is calculated until the moment the short waves break, not when the infragravity wave break as stated here (as van Dongeren et al. derived eq. (3) based on shoaling zone data)?

  We agree with the reviewer's comment and are now calculating infragravity wave amplitude until short wave break. We explicitly state this in lines 327-328 in the revised manuscript. This is also reflected in our new Figure 13 of the infragravity and short wave amplitude profile.

- Lines 296-298: I guess the short-wave height is shoaled according to linear wave theory until H=gamma*h? It would be good to say it explicitly.

  The reviewer's statement is correct: the short-wave height is shoaled using linear wave theory until H=gamma*h. This is now added as part of the new rewritten paragraphs on the model (see lines 322-328 in revised manuscript).

- Line 303: I find the formulation confusing. I do not expect that the model described in the paragraph above is needed to determine that 25 s waves are in shallow water for h<15m (which depends only on the dispersion relation). Also, how is this info used to determine that the infragravity waves have a 0.87 m amplitude at this depth? Does it mean that eq. (2) is used until the point where the waves enter shallow water (let's say h/L=0.05)? In that case, does it mean that van Dongeren et al. relation is used only shoreward of that point? (while to the best of my knowledge this empirical relation was not derived using shallow water data - see also comment #11)

We thank the reviewer for this comment and agree that line 303 was confusing. Now that we are including a figure (Figure 13) with infragravity wave amplitude profile, some of these descriptions are no longer necessary, including line 303, which we now do not include in the revised manuscript. We now include a more streamlined description of our results, focusing on the differences in infragravity wave amplitudes (see lines 331-334 in the revised manuscript). Regarding the application of the van Dongeren et al. relation, we thank the reviewer again for pointing out our mistake in our model. As we stated in our response to #12, we have now corrected this mistake and are now using the van Dongeren et al. relation in our model for the entire shoaling region, starting at kh=1.1 (largest kh in van Dongeren et al.). We state this in lines 315-316 in the revised manuscript. As we stated previously, we are now also using an improved formulation in place of our original equation 2. The new formulation (equation 3 in the revised manuscript) is valid for a wide range of water depths, as long as the group length is great compared to depth, compared to the original formulation, which is valid for deepwater.

- Lines 305 and 306: Different alpha's are used for the 10s and 25s short wave cases. I understand that these depend on the beta-values, but how are these beta's calculated? Do they differ only because of differences in breaking depth for the 10 and 25s short- wave periods? Or is omega, the IG wave frequency, changing as well when the carrier frequency is changing? If that's the case, how is it changing/what is the assumption to calculate this frequency?

  In our revised manuscript, we now state the parameters used to calculate beta immediately after we've introduced the definition of beta (see lines 312-314 in the revised manuscript). To answer the reviewer's question, we assume 12 waves per wave group in both 10s and 25s short-wave cases. Hence, omega is also changing.

- Line 306: an infragravity wave of amplitude 2.3 m (so 4.6 m high) in 5.8 m depth seems very large. Is it a typo or is there a problem with the model?

  We thank the reviewer for this comment. As we stated in our responses to #12, and also 2 bullets above, we recognized the mistake in our previous model formulation (i.e. using van Dongeren et al. relation only in shallow water), and have corrected this now thanks to the author's comment (i.e. we are now using van Dongeren et al. relation for the entire shoaling region, starting at kh=1.1, which we calculated from van Dongeren et al. data). We now compute maximum infragravity wave heights of roughly 0.40 m and 0.39 m respectively for 25 s and 10 s carrier waves (the difference between the two cases is in energy dissipation, rather than maximum infragravity wave amplitudes, see 335-352 in revised manuscript). We have compared our modeling results to published field data. For example, on a similar beach, in the same state, Fiedler et al. 2018 (Fig. 2) show an infragravity wave height of about 0.6 m at roughly 6 m water depth, at the start of carrying wave breaking. This is for a case (refer to their Fig. 1) with approximately 10 s period, 3 m wave

height offshore incident waves. When we ran their case, we computed infragravity wave height of about 0.55 m at roughly 6.4 m water depth, also at start of carrier wave breaking.

- Line 308: It seems unlikely that waves that are 5m high offshore break at a depth of 1.8 m (I would expect the breaking depth to be closer to 6-7m). Is this a typo?

  This was indeed a typo. The breaking depths are 9.85 m and 3.3 m for carrier waves of 10 s and 25 s, respectively. We don't describe the breaking depths in the revised manuscript, but this can be inferred from Figure 13 in our amplitude profiles.

14. I have similar comments/questions on the paragraph discussing the dissipation patterns based on the beta_H-value

- How is beta_H calculated in both cases? Again, beta_H depends on the infragravity wave period (or frequency), not on the short-wave period. So some explanations are missing.

  We thank the reviewer for catching this as well. As we described in a previous comment, these calculations are based on omega (infragravity wave frequency). We calculated this by assuming 12 waves per group. We added this information in line 313 in the revised manuscript, immediately after we have introduced the beta (not beta_H) parameter, where omega is first defined.

- According to the definition of beta_H, a larger IG wave height H means a smaller value of bet`a_H, and thus a smaller R (meaning more dissipation). So unless other parameters are varying (such as the long wave period), I do not understand how it can be concluded that the case with larger infragrvaity waves is dissipating the least (lines 322-324).

  In our revised model, we compute similar IG wave height H for both 25 s and 10 s carrier waves. However, we compute a much higher beta_H (and therefore much higher reflection coefficient) for 25 s carrier waves. Omega (infragravity wave frequency) is much lower for this case. This is described in lines 343-344 in the revised manuscript.

15. Finally, the link between dissipation and run-up does not seem that straightforward to me as we are comparing the runup of waves that have different incoming heights (and maybe period?). A wave that dissipates more can still lead to a higher runup than a smaller that would have dissipated less...

   As we stated in our responses to some of the previous comments, in our revised model, we are computing similar nearshore infragravity wave heights for 10 s and 25 s carrier waves, but much larger reflection coefficient for 25 s carrier waves. So it is now much

clearer to see that a wave with similar nearshore height but much less dissipation would lead to higher runup.

16. Lines 388-389: "This conclusion is supported by far offshore sensors which did not detect the incoming waves but did detect the returning ones...": The formulation suggests that the authors were able to discriminate between incoming and reflected waves -> reformulate?

We agree with the reviewer that we cannot discriminate between incoming and reflected waves at the DART sensors. We have reworded this sentence to "This explanation is supported by far offshore bottom sensors, which detected large waves hours after the first large runup vents were observed onshore, suggesting that the reflected waves were larger than the incoming ones." See lines 408-409 in the revised manuscript.

17. It should be clear in the conclusion that the predictor developed in this study is (likely to be) highly site-specific (depends on a dimensional parameter, involves a proportionality factor that already varies strongly along the considered stretch of coast...).

We have improved our predictive model in our revised manuscript. Both x and y parameters are now nondimensionalized, and the proportionality factors are now close to unity across 4 of the 5 station pairs. This makes way for the possibility of collapsing the 4 gages into one relationship (we mention this as well in the revised manuscript, see line 380-381 in the revised manuscript, though we have not done the collapse).

**Technical corrections**

- "very low frequency" is usually used to describe motions happening at a much lower frequency than the long swells described in this paper, which I find a bit confusing.

  We agree on the confusion caused by the term "very low frequency" and have rephrased them as "low frequency swell." (See lines 223 and 362 in the revised manuscript).

- Line 228: shouldn't it be figures 8 and 9 instead of figures 9 and 10?

  The reviewer is correct. We have made this change in our revised manuscript (see line 256-257 in the revised manuscript).

- Line 346: I guess that sqrt(m_{-3}f_{m}^3) should be sqrt(m_{-3}) here?

  The reviewer is correct. However, we no longer use this statement in the revised manuscript. Instead, we are describing the comparison between different negative

**Anonymous Referee #2**

The paper here does a lot of work to bring together all of the data for some extreme wave runup events, and then performs some analysis. The work with the data is very good, but the analysis is not.

We thank the reviewer for their review and comments. Following the reviewer's comment and suggestions, we have made great improvements to our paper, especially on the model and the predictive method. Please see below for our responses to specific comments.

The best work done by the authors is represented in Figure 14 where they show how low-frequency weighted moments of offshore waves relate to onshore RMS fluctuations of onshore tide gauges. This is useful, but not entirely clear as it would have been better to compare results using the m-3 moment to results using other spectral moments. sqrt(m-3) is linearly proportional to wave height. The authors suggest results but don't actually show anything definitively. The proportionality is also dimensional, which is a real problem. If the mechanism proposed here is true, then it should be able to be reduced to a dimensionless fit, but the authors state that this does not work. Why? This question must be answered.

We thank the reviewer and agree on these points. We have followed the reviewer's suggestion and improved our predictive method by 1) nondimensionalizing our water level RMS and negative moments, 2) presenting comparisons across several choices of negative moments, and 3) offering two versions of our predictive relationships, including one with more physically based nondimensionalization, but a somewhat worse fit. Much of our section on the predictive method has been rewritten with improved clarity. Please see lines 361-395, and Figure 15 and 16 in the revised manuscript.

A second issue, not addressed here, is that all tide gauges have stilling wells that deliberately filter out higher frequency water level fluctuations. I am not sure of the frequency response, but am certain that higher frequency components will be more damped than lower frequency components. For this reason the lower frequency components will be disproportionaly represented in the signals. How do the authors account for this?

The reviewer is correct that higher frequency water level fluctuations are filtered out. The tide gages samples at 1 Hz and report averages over 1 min. However, the extreme runup events captured on video show that their frequency is much longer than the frequencies of incident waves. As we stated in the paper, the frequency of the extreme runup events in the videos are close to the roughly 5 minutes peaks from the tide gages. We made a clarification on how the sampling is done at the tide gages (see line 111 in the revised manuscript).

I have harsh words for the model presented here. The authors claim it is simple, but do not give anywhere near enough information for a reviewer or reader to be able to evaluate it or reproduce it. I can't evaluate the details of what this model is or how it was produced, so can't evaluate its appropriateness or accuracy or even all assumptions included in the model. I could not reproduce it even though I am familiar with this area. It also makes impossible predictions. A carrier wave amplitude of 2.5m at 25s is stated to reach an infragravity amplitude of 2.3m by 5.8m depth. This is wrong. We know that it doesn't happen because it would be observed instantly and would be the overwhelming feature. If the phenomenon were correct, it would also have to occur for low amplitude shorter period waves that occur all the time and it doesn't. As one final note, the authors refer to the wave amplitude, but don't even say whether this refers to the amplitude of the RMS wave, the significant wave, or something else. This model needs work.

We appreciate the reviewer's comment. Following the reviewer's comment and suggestions, as well as those of the first reviewer, we have made major improvements to our model. First, the other reviewer correctly pointed out a mistake in our application of the van Dongeren et al. 2007 formulation. We previously applied this formulation only in the shallow water region. However, as the first reviewer correctly pointed out, this formulation should be applied across a larger region. We have made the corrections, and our model now no longer produce infragravity wave amplitude of 2.3 m, as we previously had. Now, the largest infragravity wave amplitude in our amplitude profile, across both 25 s and 10 s carrier waves, is 0.4 m. The difference between the infragravity waves associated with 25 s carrier waves and 10 s carrier waves is primarily in energy dissipation, where we calculate significantly less energy dissipation for the infragravity waves associated with 25 s carrier waves. In addition, we have now entirely rewritten all the paragraphs on the model. We are now tying the formulation of the model naturally with explanation of the theory. We are also now showing every equation used in the model, as well as justifications for all the input parameters. Furthermore, we have now provided a figure of infragravity and carrier wave amplitude profiles from the results of our model. Please see lines 299-352, and Figure 13 in the revised manuscript for the rewritten model description and modeling analysis, as well as the newly added figure. Lastly, we wish to add a bit of comparison of our model against field data, which we have also added in a response to the first reviewer. On a similar beach, in the same state, Fiedler et al. 2018 (Fig. 2) show an infragravity wave height of about 0.6 m at roughly 6 m water depth, at the start of carrying wave breaking. This is for a case (refer to their Fig. 1) with approximately 10 s period, 3 m wave height offshore incident waves. When we ran their case, we computed infragravity wave height of about 0.55 m at roughly 6.4 m water depth, also at start of carrier wave breaking.

---

## Author Response (AR3)

First we would like to thank both reviewers for taking their time to assess our manuscript and to provide their helpful comments. Below we address the comments from reviewer #1. We thank reviewer #2 for their reading of the paper which appears to have resulted in no further comments.

Overall the manuscript has significantly improved. Most of my comments have been appropriately addressed in the revised manuscript. The remaining comments I have are all related to Section 6.2 in which the 'simple' IG model is presented and used as an attempt to explain the large run-up events. The description of the wave model itself is now much clearer, but I still have issues with the last part that links the model output ($H_{ig}$ at the breakpoint) to beta_H and ultimately to the run-up.

We thank the reviewer for their thorough assessment of Section 6.2 and their excellent suggestions. With the help of their comments, we were able to spot written mistakes in our values of beta_H and our descriptions of beta_H calculations in the previous manuscript, which we have now corrected in the revision (details in below responses to specific comments). We have also followed the other suggestions. For response to each specific comment, please see below.

• First of all, as also recognized by the authors, Beta_H is defined in van Dongeren et al. using the IG wave height "near the shoreline". Although I recognize it is quite a vague definition, calculating beta_H from IG wave characteristics at a depth of 9-13 m depth as done in the present manuscript does not seem appropriate, so I am not sure that it is a good idea to use it as such for a quantitave

This was a written mistake and we thank the reviewer for bringing it to our attention. We in fact calculate the H (IG wave height near the shoreline) in beta_H using an iterative method, starting with the maximum IG wave height $H_{max}$ (at breaking point, 9-13 m depth), but eventually arriving at a H such that $H = RH_{max}$, where R is the reflection coefficient via van Dongeren et al. Please see lines 351-354 in the revised manuscript for a detailed, corrected description. This is how we have always calculated it, so the results haven't changed (except for the written mistakes where we wrote values of R in place of beta_H, see below).

• Defining the IG wave period as 12 times the incoming wave period as a general rule of thumb (applicable for different wave periods) seems rather arbitrary. I realize that the authors need to make a choice here but I think they should comment a little bit more on it (or recognize explicitly the limitations of this choice/possible implications).

We agree with this assessment and have clarified our choices and their limitations in the revised manuscript. Please see lines 321-323.

• I could not find back the values of beta_H given by the authors based on the info given in the text. Actually, based on the definition of beta_H and the fact that the IG period is defined as 12 times the incident wave period, I would expect that [beta_H for T=25s]/[beta_H for T=10s]=2.5 (as the IG wave height is approx. the same in both cases), which does not appear to be the case.

We have a written mistake here. The values of 0.49 and 0.89 are reflection coefficients, not beta_H as we have written in the previous manuscript. We have corrected this in the revised manuscript (see lines 354-356). To further clarify, [beta_H for T=25s]/[beta_H for T=10s] is actually not 2.5 as the values of H (infragravity wave height) in the equation are also different for the two cases (see response to the first specific comment, above). Apologies for the written mistake on the description of the beta_H calculations previously.

• Assuming I made a mistake and that the values of beta_H are indeed 0.49 and 0.89 as stated in the manuscript, the associated reflection coefficients according to van Dongeren would then be 0.15 and 0.5 (not given in the manuscript), which means that in both cases quite a lot of

dissipation takes place. So I do not think the authors can claim lines 349-350 that the IG waves for a 25s carrier wave show a "low dissipation" for instance.

As we stated above, the reflection coefficients (R) are actually 0.49 and 0.89 for the 10 s and 25 s carrier waves cases, respectively. We also no longer use the word 'low dissipation' and now simply state the R values.

• I do not understand the reasoning lines 347-349 leading to the conclusion that the energy available for run-up is 3.5 times larger when the incident waves are 25 s instead of 10 s, so I cannot judge of its validity.

Actually, I am wondering if the authors are not trying to push too far their simple modelling approach (i.e. not only to estimate Hig but also the runup induced by these long waves). An alternative would be to use a numerical wave-resolving model which could be used over the same simplified bathymetry for waves of same Hs but different peak periods (assuming a given spectral shape). Anyway, if the authors decide to stick to their simplified approach the points raised above should be addressed.

We agree that making inferences on runup based on our simple model is somewhat reaching. In the revised manuscript, we now simply state the wave height and wave energy comparisons close to shore between the 10 s and 25 s carrier wave periods. Please see lines 354-357 in the revised manuscript.

• Lines 349-352: Overall I am not sure we can really support the statement of dissipation based on the videos. In particular I do not think that the fact that the run-up front slows down (or does not slow down) on the video recordings is a good indication of wave dissipation (or absence thereof). I would say that it depends mostly on the timing with respect to the run-up/run-down cycle.

We agree with the reviewer and removed our statements on dissipation based on the videos.

Other comment:
Line 262: the authors need to give some more details on the "simple shelf geometry" they use in their calculations to make their results reproducible.

We have added a detailed description of the simple shelf geometry we have used in lines 254-257 of the revised manuscript.

---

## Author Response (AR4)

We thank the reviewer once again for their time reviewing our manuscript and offering insightful comments. We hope that our minor revision as recommended by the reviewer address the issue they raised. Please see below for a more specific response.

I am glad that my comments helped identifying a number of mistakes/typos in the revised manuscript. In the new version, the authors explain more extensively several aspects of their methodology. I however found some of the new elements, and in particular the part explaining how the nearshore IG wave heights (and reflection coefficients) are estimated, very confusing. I do not understand how the iterative process described in a few lines in the new manuscript is conducted (i.e. how do we make H converge) but, even more problematic in my view, I do not understand at all the underlying reasoning (e.g., why defining the new H as R*Hmax, with R the reflection coefficient?). Maybe I am missing something but to me (in its current version) it does not make sense at all. This needs to be addressed before publication.

We understand and agree that the explanation on method of obtaining IG wave height was confusing. In the new revision, we explain how H = RHmax is used, what is assumed, and why it is needed. We also made clearer the description of how H is solved iteratively and what we mean by converging. Please see lines 351-358 in the new revision.

---

## Author Response (AR5)

Unfortunately, the authors' explanations on the calculation of the nearshore IG wave height did not alleviate my concerns. I now understand how the authors conducted their calculations, but I still do not understand the underlying reasoning. I am actually now convinced that it is incorrect.

[... the paper ] still contains some incorrect reasoning which in my view needs to be corrected (or removed) before publication.

[...]

The first part of the idealized modelling exercise leads to an estimate of the incoming IG wave height at the short wave breakpoint (called Hmax in the manuscript), which is located in about 10 m depth for the conditions considered here. This I can follow and it makes sense to me.

What does not make sense to me is how the authors use this Hmax (defined in ~10m depth) and a reflection coefficient R defined near the shoreline (so much more onshore, let's say in ~1m depth) to define the incoming wave height near the shoreline H, i.e. why it would make sense to assume that H=R*Hmax.

The reflection coefficient R is a function of the incoming and reflected wave heights at the location where this reflection coefficient is defined, in that case near the shoreline. While I agree that H in 1m depth will somehow depend on the wave height Hmax more offshore, there is no straightforward relation between the two as far as I know, and the reflection coefficient R in ~1m depth cannot be used to express how the incoming wave height decays between 10m and 1m depth.

We thank the author for taking their time to review our manuscript again. We agree that the calculations of the wave height near the shoreline, H, can be improved and have done so in this revision. In this revision, we shoaled the infragravity wave as a free wave (i.e. with Green's Law) shoreward from carrier wave breakpoint - where we ended in the previous version of the manuscript - to the infragravity wave breakpoint. We then calculated the breaking profile of the infragravity wave from its breakpoint to the shoreline using the energy equation for linear long waves, which accounts for energy dissipation due to wave breaking. This method is also used in von Dongeren et al. (2007), the reference from which we've used two key equations in the model. We then take H as the infragravity wave height at the shoreline, and use this H to find our reflection coefficient, R, in the same manner as done by von Dongeren (2007). Although the physics of reflection is not included, the model should nonetheless predict, in a relative sense when comparing the two cases, the wave height value near the shoreline. Using this improved method, we find for our key results that H = 0.88 m and H = 0.52 m for T = 25 s and T = 10 s, and R = 0.79 and R = 0.42 for T = 25 s and T = 10 s. Note that in the revised manuscript, we reported these values as the amplitude, a_ig = 1/2H rather than H to be consistent with the other text. These values turned out to be comparable to the previous values of H = 0.71 m and H = 0.38 m for T = 25 s and T = 10 s, and R = 0.89 and R = 0.49 for T = 25 s and T = 10 s, and thus the key points of this section were not changed. Furthermore, we have made a naming change of H to H_ig and H_rms to H_rms,ig, to make clear that these are variables relating to the infragravity wave and avoiding confusion with incident waves. We have also corrected a mistake in the previous version of the manuscript as the wave height available for runup should be RH instead of just H. In addition, we have rearranged the model description section to read from offshore to onshore. Aside from this rearrangement, the changes in this revision are in lines 342 to 373.

---

## Author Response (AR6)

The reviewers has evaluated positively the latest version of your manuscript, which I consider now publishable subject to minor revisions (review by the editor).

We thank the reviewer and editor again for their time in reviewing our revised manuscript and we appreciate the recommendation that the manuscript is now publishable with minor revisions. In the new revised manuscript, we have addressed the comments below.

It is important that you consider the following suggestions of the reviewer.

• Calculation of infragravity wave height over the profile.
This part is now publishable provided that the authors add a few sentences to further highlight the limitations of their approach. Besides the use of a very simple model to describe IG shoaling for instance, other important limitations include the fact that the IG wave period, which is a key parameter in several stages in the modeling exercise, is arbitrarily estimated as 12 times the short wave period, and the fact that full modulation of the wave field is assumed (so amplitude varying between H and 0) which means that the IG wave forcing is (severely) overestimated.

We agree and have added a paragraph dedicated to highlighting these limitations in lines 360-364 of the new manuscript.

• Calculation and use of the reflection coefficient.
The link between reflected IG wave height (defined as R*Hig) and runup (explained around line 405 of the 'track-changed' manuscript) is not obvious. The authors should either add references to back up their claim with references, or to simply remove the part of the manuscript discussing the reflection coefficient (and its use as measure for run-up) which is not convincing (as is) and not key to the story.

We added three well-cited references to support the link between wave reflection and runup, in particular in the infragravity frequency. We also further clarify that this link is due to the oscillations from standing waves associated with wave reflection. These are added in lines 375-379 in the revised manuscript.

As a side note (only relevant if the authors decide to keep the part involving the reflection coefficient), the authors derived their own estimate of the relation between R and beta_H based on van Dongeren et al. (2007)'s figure (R=0.5*beta_H) because according to them this relationship is not given in the paper. That's incorrect: R is explicitly defined in van Dongeren's paper as R=0.2*pi*beta_H^2.

We believe that this relationship (R=0.2*pi*beta_H^2) is actually for short waves from a different work (Battjes 1974) rather than for the R vs beta_H data in van Dongeren et al. (2007). van Dongeren et al. (2007) writes: "For short waves, Battjes [1974] found a relation between the reflection coefficient at the shoreline R and the surf similarity parameter, which can be rewritten using equation (7) as R=0.1*xi^2=0.2*pi*beta_H^2. This relationship (solid line in Figure 3) appears to also apply to low-frequency waves, albeit that there is considerable scatter." It can also be seen from their Figure 3 that the relationship considerably overestimates the data of van Dongeren (2007) at higher beta_H (higher than beta_H~1). Nevertheless, we removed the phrase "Although not explicitly stated in van Dongeren et al. (2007)" and simply states that we use R=0.5*beta_H based on the results of van Dongeren et al. (2007).

• The revised part of the manuscript contains several typos (units are not provided, spaces are missing, etc.) that need to be fixed.
We have fixed typos in the new manuscript.